# Interpretable language modeling via induction-head ngram models

## Abstract

Recent large language models (LLMs) have excelled across a wide range of tasks, but their use in high-stakes and compute-limited settings has intensified the demand for interpretability and efficiency. We address this need by proposing Induction-head ngram models (Induction-Gram), a method that builds an efficient, interpretable LM by bolstering modern ngram models with a hand-engineered "induction head". This induction head uses a custom neural similarity metric to efficiently search the model's input context for potential next-word completions. This process enables Induction-Gram to provide ngram-level grounding for each generated token. Moreover, experiments show that this simple method significantly improves next-word prediction over baseline interpretable models (up to $26\%p$) and can be used to speed up LLM inference for large models through speculative decoding. We further study Induction-Gram in a natural-language neuroscience setting, where the goal is to predict the next fMRI response in a sequence. It again provides a significant improvement over interpretable models ($20\%$ relative increase in the correlation of predicted fMRI responses), potentially enabling deeper scientific investigation of language selectivity in the brain.

## 1 Introduction

Large language models (LLMs) have demonstrated remarkable predictive performance across a growing range of diverse tasks (Brown et al., 2020; OpenAI, 2023; Dubey et al., 2024). However, their proliferation has led to two burgeoning problems. First, LLMs have become increasingly difficult to interpret, often leading to them being characterized as black boxes and debilitating their use in high-stakes applications such as science, medicine, and policy-making (Birhane et al., 2023; Thirunavukarasu et al., 2023; Singh et al., 2024). Moreover, the use of LLMs has come under increasing scrutiny in settings where users require explanations or where models struggle with issues such as fairness (Li et al., 2023) and regulatory pressure (Meskó & Topol, 2023). Second, LLMs have grown to massive sizes, incurring enormous energy costs (Bommasani et al., 2023) and making them costly and difficult to deploy, particularly in low-compute settings (*e.g.*, edge devices).

As an alternative to LLMs, ngram models can maintain complete interpretability and are significantly more computationally efficient. While interpretable models can perform as well as black-box models in some domains (Rudin et al., 2021; Mignan & Broccardo, 2019; Ha et al., 2021), there is a considerable gap between the performance of interpretable models and black-box LLMs in next-token prediction.

To shrink this gap, we propose Induction-head ngram models (Induction-Gram), a method to build interpretable and efficient LMs by bridging ngram LMs with neural LLMs. Specifically, Induction-Gram starts with Infini-Gram, a state-of-the-art scalable ngram model (Liu et al., 2024). While effective, Infini-Gram struggles with adapting to new contexts and with matching queries that can not be found exactly within a reference dataset (*e.g.*, typos or rephrasings). To remedy these issues, Induction-Gram uses fuzzy matching within the model's context to retrieve suggestions for a next-token completion, similar to the role played by "induction heads" found in pre-trained transformer models (Olsson et al., 2022; Akyürek et al., 2024). Similarly, Induction-Gram performs matching by using a custom neural similarity metric that is trained to efficiently score two texts as similar precisely if they lead to similar next-token completions. This extension allows Induction-Gram to achieve state-of-the-art next-token prediction accuracy for an interpretable language model.

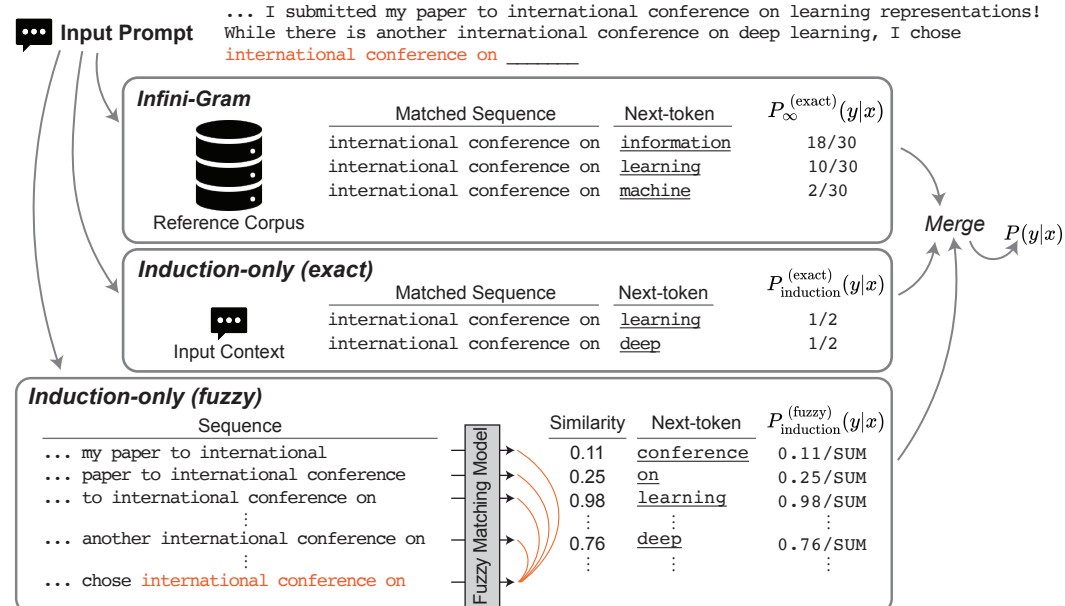

Figure 1: Overview of Induction-Gram pipeline. Induction-Gram predicts the next token by integrating an ngram model (Infini-Gram) with a constructed "induction head", that efficiently searches for potential next-token completions in the input context.

For example, when evaluating on the Pile dataset and using OpenWebText as the reference corpus, Induction-Gram improve next-token prediction accuracy by 20%p over standard Infini-Gram, shrinking the gap between interpretable models and the black-box GPT-2 model (see Table 1).

We further explore Induction-Gram in a natural-language fMRI context, where the goal is to predict the next fMRI response in a session rather than the next token in a sequence. In this setting, Induction-Gram yields a 20% improvement over the baseline interpretable model and allows for auditing how models adapt to local context. Overall, Induction-Gram constitutes a major step towards reverse-engineering mechanistically interpretable language models from modern LLMs.

## 2 RELATED WORK

**ngram language models.** Early language modeling techniques revolved around ngram models (Jurafsky & Martin, 2000; Katz, 1987), which generally stored next-token probabilities in large tables learned from data (Brants et al., 2007). While neural LLMs have generally surpassed ngram LMs, recent works have continued to improved ngram LMs, *e.g.*, by scaling up the ngram reference data (Allamanis & Sutton, 2013) and improving the ngram probability representations using suffix arrays and suffix trees (Stehouwer & van Zaanen, 2010; Kennington et al., 2012; Shareghi et al., 2015). This line of work culminated in Infini-Gram (Liu et al., 2024), which efficiently scales ngram models to massive datasets and is the starting point for our work.

**Bridging interpretable models and LLMs** Some works have studied bridging ngram models and LLMs. For example, Khandelwal et al. (2020) interpolate neural LMs with an ngram model and Li et al. (2022) train a neural model to complement an ngram model. He et al. (2023) use ngram models to speed up LLM inference via speculative decoding (He et al., 2023). Another approach builds black-box nonparametric LMs using techniques such as $k$-nearest neighbor to improve LLM predictions (Khandelwal et al., 2020; Borgeaud et al., 2022). Our Induction-Gram LM is also based on a nonparametric LM, but unlike these other works, it maintains complete interpretability during inference. In simplified settings such as text classification, some works have built fully interpretable models that bridge LLMs and ngram models (Li et al., 2017; Singh et al., 2023a) or built partially interpretable models based on approximating concepts with natural language (Yang et al., 2023a; Sun et al., 2024; Morris et al., 2023; Feng et al., 2024).

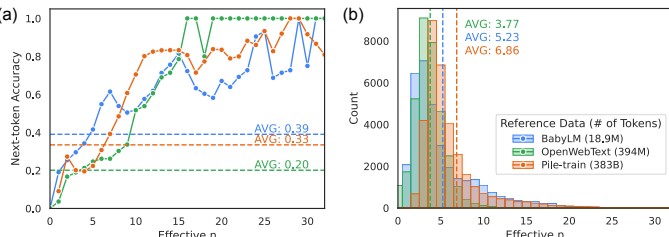

Figure 2: Performance on the BabyLM dataset with Infini-Gram built from various reference datasets. (a) Next-token prediction accuracy for each effective $n$. The dashed line indicates the average accuracy. (b) The histogram illustrates the count for each effective $n$.

In parallel, there has been a surge of recent interest in mechanistic interpretability, which seeks to understand what mechanisms are learned by transformer-based LLMs (Rai et al., 2024). This line of work identified induction heads in toy LLM models (Olsson et al., 2022) as well as large-scale pre-trained LLMs (Wang et al., 2022; Akyürek et al., 2024).

**Natural language representations in fMRI**   In recent years, predicting brain responses to natural language using LLM representations has become common in the field of language neuroscience (Jain & Huth, 2018; Wehbe et al., 2014; Schrimpf et al., 2021; Goldstein et al., 2022). This paradigm of using predictive "encoding models" to better understand how the brain processes language has been applied in a wide literature to explore to what extent syntax, semantics, or discourse drives brain activity (Wu et al., 2006; Caucheteux et al., 2021; Kauf et al., 2023; Reddy & Wehbe, 2020; Kumar et al., 2022; Oota et al., 2022; Tuckute et al., 2023; Benara et al., 2024; Antonello et al., 2024a) or to understand the cortical organization of language timescales (Jain et al., 2020; Chen et al., 2023a). Separately, many works study the behavior of humans at recalling and processing repeated text (Baddeley, 1992; Tzeng, 1973; Amlund et al., 1986; Miles et al., 2006) and relating it to LLMs (Vaidya et al., 2023; Pink et al., 2024). Our work bridges these two areas, exploring whether we can explicitly understand the cortical representations involved in recalling context by predicting brain responses using Induction-Gram.

## 3   METHOD

We first introduce Infini-Gram, the ngram method we build on (Sec. 3.1), then introduce the efficient induction head we develop (Sec. 3.2), before we combine them to yield Induction-Gram (Sec. 3.3).

### 3.1   PRELIMINARIES: INFINI-GRAM

Given an input text sequence, Infini-Gram (Liu et al., 2024) searches a reference corpus for the examples with the longest exact suffix match to the input, then calculates the next-token distribution based on the token following each of the matches. This search is made extremely efficient by building large-scale suffix arrays that can scale to trillions of reference tokens. The length of the longest match is referred to as the *effective* $n$, with the accuracy of the estimated probabilities increasing as the *effective* $n$ becomes larger.

One limitation of Infini-Gram is that finding exact matches in the reference corpus becomes challenging when there is a distribution shift between the input context and the reference corpus. For instance, when evaluating on the BabyLM[1] test dataset, Infini-Gram built on larger corpora, such as OpenWebText (Gokaslan & Cohen, 2019), shows lower performance and, on average, has fewer instances of higher effective $n$ compared to the model built on the BabyLM dataset (Fig. 2). With far larger corpora like Pile-train (Gao et al., 2020), Infini-Gram is able to increase the number of instances with a high effective $n$, resulting in improved performance. However, the Infini-Gram built on BabyLM, which contains only 0.005% of the tokens found in Pile-train, still achieves the highest performance. This highlights the difficulty Infini-Gram faces when there is a substantial gap between the reference corpus and the input prompt, making it hard to find matching cases with a large effective $n$. We propose to address this limitation with Induction-Gram.

---

[1] https://babylm.github.io/

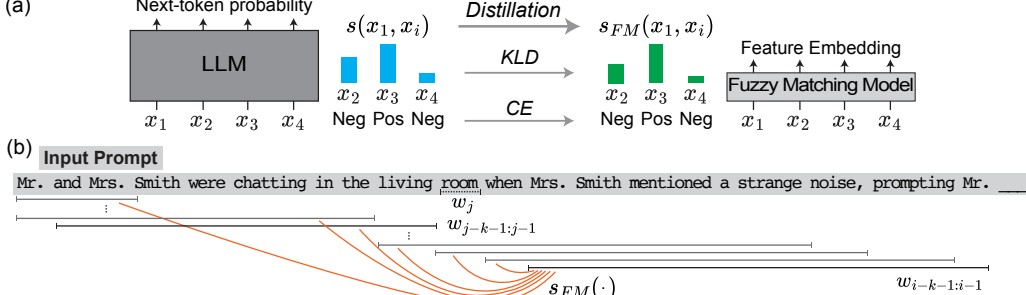

Figure 3: (a) Overview of training Fuzzy Matching Model via knowledge distillation from pretrained LLM. (b) Calculation of similarity between sequences within input prompt to predict the next token.

## 3.2 BUILDING AN EFFICIENT INDUCTION HEAD

LLMs are well-known for their ability to perform in-context learning, effectively capturing the distribution of input context. In pre-trained LLMs, the induction head has been found to play a crucial role in in-context learning (Olsson et al., 2022; Akyürek et al., 2024; Wang et al., 2022), which refers to attention patterns in LLMs that identify recurring sequences in prior context and use them to predict the next token (*e.g.*, [A][B] ... [A] → [B]). To replicate this behavior, we propose to construct an induction head based on ngrams to aid in next-token prediction. Building this induction head is similar to applying the Infini-Gram algorithm restricted only to the input context: it treats the end of the context as the query and searches for the best match within the context. After finding the best match, the induction head takes the token following the match as the next-token prediction.

**What constitutes a "good match" for our induction head?** When finding an ngram-level match within the context, exact matching can be overly restrictive, as minor rephrasings or typos may derail an otherwise useful match. Consequently, we adopt fuzzy matching instead of exact matching by assessing the similarity between sequences. While similarity can be defined in many ways, in building an induction head we desire two texts to be similar if they yield similar next-token distributions. To quantify this, we define the similarity between two sequences, $x_1$ and $x_2$, for fuzzy matching using Jensen–Shannon divergence (JSD), as follows:

$$s(x_1, x_2) = \exp\left(-\text{JSD}\left(P_{\text{next}}(x_1), P_{\text{next}}(x_2)\right)\right), \quad (1)$$

where $P_{\text{next}}(\cdot)$ is the estimated next-token probability distribution for a given sequence.

**Computing $s$ efficiently** One approach for computing $s$ would be to use a pre-trained LLM to obtain $P_{\text{next}}$, but this can be computationally expensive. Instead, we develop a small Fuzzy Matching Model, which consists of 3 or 4 transformer layers and is trained via knowledge distillation from existing LLMs. This model is designed to output feature embeddings that facilitate the calculation of next token probabilities for similarity assessments. With Fuzzy Matching Model, the similarity between $x_1$ and $x_2$, whose feature embeddings from the model are $e_1$ and $e_2$, is obtained as follows:

$$s_{\text{FM}}(x_1, x_2) = \exp\left(-\left(1 - \text{CosineSim}\left(e_1, e_2\right)\right)/T\right), \quad (2)$$

where $T$ is a temperature, which is set to 0.1. The Fuzzy Matching Model is trained using a combination of Cross Entropy (CE) loss and reverse Kullback-Leibler divergence (KLD) loss (Fig. 3(a)). Within each training batch, we create similarity pairs from randomly sampled sequences with an LLM. The CE loss aids in identifying the most similar pairs. The reverse KLD loss encourages the model to align with the distribution of similarity, emphasizing the importance of accurately estimating the overall similarity while ensuring that the closest pairs receive high similarity scores and the distant pairs receive lower similarity scores. Further details can be found in Appendix A.1.

**Predicting the next token** Given the similarity scoring function $s_{\text{FM}}$, we can build an induction head that yields the predicted next-token probability distribution $P_{\text{induction}}$ given an input sequence $x$. To do so, we find each match for the end of $x$, $w_{:i-1}$, using a sliding window of size $k$ (Fig. 3(b)). We then count the occurrence of each token $w_i$, among vocabulary set $\mathcal{V}$, following each match in

the input sequence, and then normalize to obtain the next-token probability:

$$P_{\text{induction}}^{\text{(fuzzy)}}(w_{:i-1}w_i|x) = \frac{c_{\text{fuzzy}}(w_{i-k-1:i-1}w_i|x)}{\sum_{w_j \in \mathcal{V}} c_{\text{fuzzy}}(w_{i-k-1:i-1}w_j|x)} \quad (3)$$

$$\text{where } c_{\text{fuzzy}}(w_{i-k-1:i-1}w_i|x) = \sum_{w_{j-k-1:j} \subset x} \mathbb{1}_{w_j=w_i} s_{\text{FM}}\left(w_{j-k-1:j-1}, w_{i-k-1:i-1}\right). \quad (4)$$

This similarity score serves as a floating count for the next token. In cases where the sequences $x_1$ and $x_2$ are exactly matched, as in the case of Infini-Gram, we have $s_{\text{FM}}(x_1, x_2) = 1$, which is equivalent to increasing the count by one. The window size $k$ specifies the number of tokens to be considered in fuzzy matching.

### 3.3 INDUCTION-HEAD NGRAM MODELS: PUTTING IT ALL TOGETHER

To build our final Induction-Gram model (Eq. (5)), we integrate our induction head with the baseline Infini-Gram model, which uses exact ngram matching:

$$P(y|x) = \begin{cases} P_{\infty}^{\text{(exact)}}(y|x) & n_{\infty} > n_x \text{ and } n_{\infty} > \tau, \\ P_{\text{induction}}^{\text{(exact)}}(y|x) & n_x \geq n_{\infty} \text{ and } n_x > \tau, \\ P_{\text{induction}}^{\text{(fuzzy)}}(y|x) & \text{Otherwise,} \end{cases} \quad (5)$$

where $n_{\infty}$ and $n_x$ are the effective $n$ when matching from a reference corpus or the input context, respectively. When these values are low, fuzzy matching is employed to compensate for the limited    FIX
effective $n$. When the effective $n$ values from both the input context and reference corpus are equal, priority is given to the input context estimate. $\tau$ is a hyperparameter that selects how often to use exact matching rather than fuzzy matching; we set $\tau$ to 8 and 9 for GPT-2 and LLaMa-2 tokenizers, respectively, using cross-validation test (details in Appendix A.2).

While we describe Induction-Gram for text, it can be applied to predicting tokens in sequences more generally; Sec. 5.1 describes how to use Induction-Gram in a natural-language fMRI setting.

## 4 LANGUAGE MODELING RESULTS

### 4.1 EXPERIMENTAL SETUP

**Datasets**   We use 4 text datasets for evaluation: BabyLM[2] (Warstadt et al., 2023), OpenWeb-   FIX
Text (Gokaslan & Cohen, 2019), Pile (Gao et al., 2020), and FineWeb ((Penedo et al., 2024); `sample-10BT` subset), using some as the reference corpus and some as test datasets (Table 1). When testing, we report performance on 100k sequences randomly sampled with a context length of 1024 and a stride of 512 (Liu et al., 2024; Khandelwal et al., 2020).[3] In our speculative decoding experiments, we utilize 1024 tokens from the beginning of each document as a prefix prompt. Six prompts are employed with the BabyLM dataset, while 100 randomly sampled prompts are used for the FineWeb and Pile datasets.

**Metrics**   We evaluate our method in terms of both the accuracy and efficiency of next-token prediction. We measure accuracy as whether the top-predicted token was the correct token.[4] For efficiency, we compare the inference time for speculative decoding (Leviathan et al., 2023; Chen et al., 2023b) when using Induction-only (fuzzy) as the draft model.

### 4.2 IMPROVING NEXT-TOKEN PREDICTION ACCURACY WITH CONTEXTUALIZATION

**Prediction improvements from in-context matching**   Induction-only (exact) relies solely on the input context to predict the next token (limited to 1024 tokens in our evaluation). Table 1 shows

---

[2] https://babylm.github.io/

[3] The BabyLM test set results in less than 100k sequences, instead yielding about 32k and 34k cases for the GPT-2 and LLaMA-2 tokenizers, respectively.

[4] We do not compute perplexity, as the sparse next-token predictions from ngram models can frequently assign the top token a probability of zero, skewing the perplexity to extreme values.

Table 1: Next-token prediction accuracy (%) for **Induction-Gram** compared to baseline methods. The gray shade represents the alignment between the reference corpus and the test dataset.

| Reference Corpus | | Model | Test Dataset | | |
|---|---|---|---|---|---|
| Type | # of Tokens | | BabyLM-test | FineWeb | Pile-val |
| **Tokenizer: GPT-2** | | | | | |
| - | - | Induction-only (exact) | 36.7 | 17.2 | 37.0 |
| - | - | Induction-only (fuzzy) | 41.1 | 25.2 | 38.7 |
| BabyLM-dev | 17.4M | Infini-Gram | 37.6 | 14.7 | 16.0 |
| | | **Induction-Gram** | 42.2 (+4.6) | 25.3 (+10.6) | 40.0 (+24.0) |
| Pile-val | 383M | Infini-Gram | 16.6 | 20.1 | - |
| | | **Induction-Gram** | 41.5 (+24.9) | 25.5 (+5.4) | - |
| OpenWebText | 9.04B | Infini-Gram | 16.7 | 25.5 | 22.7 |
| | | **Induction-Gram** | 41.8 (+25.1) | 27.2 (+1.7) | 42.7 (+20.0) |
| Unknown | ∼10B | LLM (GPT-2) | 46.9 | 39.0 | 52.3 |
| **Tokenizer: LLaMA-2** | | | | | |
| - | - | Induction-only (exact) | 37.0 | 19.6 | 32.6 |
| - | - | Induction-only (fuzzy) | 42.7 | 28.3 | 38.5 |
| BabyLM-dev | 18.9M | Infini-Gram | 39.0 | 17.1 | 13.2 |
| | | **Induction-Gram** | 43.1 (+4.1) | 28.6 (+11.5) | 39.6 (+26.4) |
| Pile-val | 394M | Infini-Gram | 19.0 | 24.1 | - |
| | | **Induction-Gram** | 42.9 (+23.9) | 28.4 (+4.3) | - |
| OpenWebText | 10.3B | Infini-Gram | 20.1 | 29.5 | 27.1 |
| | | **Induction-Gram** | 43.2 (+23.1) | 30.3 (+0.8) | 42.1 (+15.0) |
| Pile-train | 383B | Infini-Gram | 33.5 | 39.3 | 49.2 |
| | | **Induction-Gram** | 49.4 (+15.9) | 38.0 (-1.3) | 50.3 (+1.1) |
| Unknown | ∼2T | LLM (LLaMA2-7B) | 62.2 | 57.1 | 64.4 |

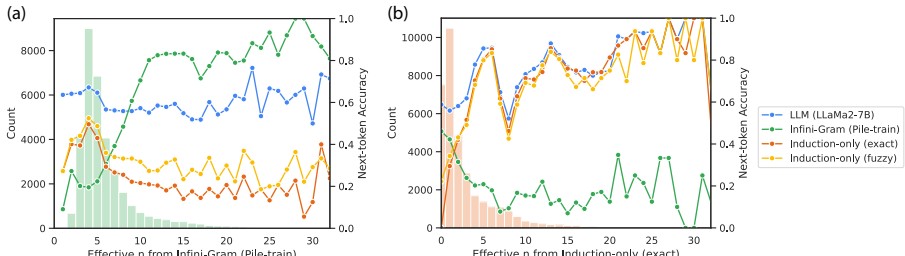

Figure 4: Comparison of next token prediction accuracy on BabyLM-test dataset, depending on effective $n$ from (b) Infini-Gram and (b) Induction-only (exact). LLaMA-2 tokenizer is used.

that, despite this, it outperforms Infini-Gram—which uses the 10B-token OpenWebText dataset as a reference corpus—by a margin of 5.5%p to 20%p on the BabyLM and Pile datasets. When Infini-Gram utilizes BabyLM-dev as the reference corpus, it achieves slightly better performance than Induction-only (exact) on the BabyLM-test set, with improvements of 0.9%p and 2.0%p for the GPT-2 and LLaMA-2 tokenizers, respectively, where the reference corpus and input context are aligned. As shown in Fig. 4(a), Infini-Gram (green) performs better in cases with a high effective $n$, even surpassing LLM (blue). However, there are significantly more cases with a low effective $n$ (histogram), where Induction-only (exact) (orange) demonstrates superior performance. This finding underscores that in-context matching reflects the input query's distribution, resulting in more accurate next-token predictions than reference matching, especially when there is a distribution shift between the reference corpus and the test input.

**Prediction improvements from Induction-Gram** Induction-only (fuzzy), using Fuzzy Matching Model, consistently outperforms Induction-only (exact) with a margin of 1.7%p to 8.7%p (Table 1). This improvement is particularly evident in cases with low effective $n$. As illustrated in Figure 4(b), the majority of cases within the input context have low effective $n$ (histogram), indicating that finding exactly matched long sequences within the limited amount of tokens is challenging. Fuzzy matching helps to provide better estimations for next-token predictions in these scenarios.

Table 2: Speed of speculative decoding (SP). Accept. denotes the acceptance rate (%). The mean and standard deviation of 3 runs are reported.

| | Draft Model | Large Model | SP | BabyLM-test | | | Pile-val | | |
|---|---|---|---|---|---|---|---|---|---|
| | | | | Accept. | Speed | | Accept. | Speed | |
| | | | | | ms/token ($\downarrow$) | Up ($\uparrow$) | | ms/token ($\downarrow$) | Up ($\uparrow$) |
| A40×1 | | LLaMA2-7B | | | 30.2±0.0 | | | 30.2±0.1 | |
| | TinyLLaMA-1.1B | LLaMA2-7B | ✓ | 78.7±0.5 | 21.3±0.0 | 1.42 | 78.3±0.1 | 21.3±0.6 | 1.42 |
| | Induction-only (fuzzy) | LLaMA2-7B | ✓ | 74.9±1.1 | 17.7±0.7 | 1.71 | 71.2±0.5 | 20.1±0.4 | 1.50 |
| | | LLaMA2-13B | | | 52.4±0.0 | | | 52.0±0.2 | |
| | TinyLLaMA-1.1B | LLaMA2-13B | ✓ | 78.2±0.0 | 26.7±0.5 | 1.96 | 77.6±0.1 | 26.3±0.5 | 1.98 |
| | Induction-only (fuzzy) | LLaMA2-13B | ✓ | 73.5±0.1 | 24.8±0.1 | 2.11 | 69.8±0.2 | 27.8±0.1 | 1.87 |
| H100×2 | | LLaMA2-13B | | | 26.4±0.1 | | | 26.3±0.4 | |
| | LLaMA2-7B | LLaMA2-13B | ✓ | 78.9±0.0 | 24.7±0.0 | 1.07 | 78.6±0.0 | 25.1±0.3 | 1.05 |
| | TinyLLaMA-1.1B | LLaMA2-13B | ✓ | 78.3±0.1 | 20.7±0.1 | 1.28 | 77.6±0.1 | 21.5±0.1 | 1.22 |
| | Induction-only (fuzzy) | LLaMA2-13B | ✓ | 73.2±0.3 | 13.3±0.2 | 1.98 | 69.9±0.1 | 14.9±0.1 | 1.77 |
| | | LLaMA2-70B | | | 71.2±0.1 | | | 71.0±0.2 | |
| | LLaMA2-7B | LLaMA2-70B | ✓ | 77.2±0.2 | 38.3±0.5 | 1.86 | 77.8±0.2 | 37.4±0.3 | 1.90 |
| | TinyLLaMA-1.1B | LLaMA2-70B | ✓ | 75.5±0.1 | 35.3±0.2 | 2.02 | 76.3±0.4 | 33.9±0.6 | 2.10 |
| | Induction-only (fuzzy) | LLaMA2-70B | ✓ | 68.5±0.6 | 31.4±0.7 | 2.27 | 66.6±0.6 | 33.3±0.6 | 2.13 |

Specifically, when the effective $n$ is less than 3, Induction-only (fuzzy) (yellow) demonstrates better performance than Induction-only (exact) (orange). Since many cases fall into this range, the overall accuracy of Induction-only (fuzzy) is higher.

The improvements achieved through the use of induction and fuzzy matching enable Induction-Gram to outperform Infini-Gram built on 383B tokens improving performance by up to 16.0%p. While expanding the reference corpus of Infini-Gram can lead to general performance gains, utilizing Induction-only (fuzzy) proves to be more efficient than increasing the data size from 10.3B to 383B tokens—a 38-fold increase. Moreover, Induction-only (fuzzy) is a complementary approach that can be applied orthogonally to Infini-Gram, regardless of the size of the reference corpus.

### 4.3 SPECULATIVE DECODING

**Experimental Details** To evaluate the efficiency of Induction-only (fuzzy), we compare the inference time for speculative decoding with TinyLLaMA[5] and LLaMA2-7B (Touvron et al., 2023). We evaluate speculative decoding by generating up to 1024 tokens, using a prefix of 1024 tokens. The speed of decoding may vary depending on the computational environment. To ensure robust evaluation across different setups, we conduct experiments in two environments: one with a single NVIDIA A40 GPU and 128 CPU cores, and another with two NVIDIA H100 GPUs and 64 CPU cores. Greedy sampling is used for token generation, and each experiment is repeated three times with different random seeds.

**Induction improves speculative decoding performance** Table 2 demonstrates the speed-up effect of speculative decoding with Induction-only (fuzzy). Induction-only (fuzzy) relies solely on the induction power derived from the input context to predict the next token, leading to lower acceptance rates compared to LLMs. Despite this, its inference speed is remarkably fast, and it often matches the predictions of large models. As a result, the speed improvement can exceed $2\times$ compared to using LLaMA2-70B alone. In most cases, Induction-only (fuzzy) achieves even greater speed gains than when using an LLM as a draft model for speculative decoding.  NEW

Additionally, we would like to note that speculative decoding with Induction-only (fuzzy) and a pretrained LLM not only accelerates the inference speed of the pretrained model but also enables explainable predictions based on the given input context. When accurate predictions can be made through interpretable methods, we utilize this process for interpretability. In more challenging cases, we rely on a larger model that, while less interpretable, delivers better performance for accurate predictions. Thus, this approach provides a balanced method that addresses both interpretability and accuracy, in addition to enhancing efficiency.

---

[5] https://huggingface.co/TinyLLaMA/TinyLLaMA-1.1B-intermediate-step-1431k-3T

## 5 FMRI RESULTS

### 5.1 EXPERIMENTAL SETUP

A central challenge in neuroscience is understanding how and where semantic concepts are represented in the brain. To meet this challenge, we follow a line of study that predicts the response of different brain voxels (i.e. small brain regions) to natural language stimuli (Huth et al., 2016; Jain & Huth, 2018). We analyze data[6] from LeBel et al. (2022) and Tang et al. (2023), which consists of fMRI responses for human subjects as they listen to 20+ hours of narrative stories from podcasts. We fit modules to predict the fMRI response (95,556 voxels) from the text that a single subject was hearing by extracting text embeddings[7]. We fit the encoding models on the training split (24 stories) and evaluate them on the test split (2 stories) using bootstrapped ridge regression. Encoding model features are extracted in various ways (described below) for each word in the input, and then interpolated to make predictions for the fMRI data that is recorded at 2-second time of repetition (TR) intervals. To model temporal delays in the fMRI signal, we also add 4 time-lagged duplicates of the input features. See extended fMRI details in Appendix A.4.

**Embedding baselines** We use Eng1000 as our primary baseline, an interpretable model developed in neuroscience literature for predicting fMRI responses from narrative stories (Huth et al., 2016). Each element in an Eng1000 embedding corresponds to a co-occurence statistic with a different word. We additionally compare to embeddings from LLaMA2-70B (Touvron et al., 2023), which achieve state-of-the-art performance in this fMRI prediction task (Antonello et al., 2024b) but are not interpretable. LLaMA embeddings are extracted using a 16-word sliding window and selecting the final-layer embedding for the final token of the input.

**fMRI induction head settings** We construct our induction head for fMRI by searching over recent text in an fMRI session and identifying previous changes in the recorded fMRI response. Specifically, to predict the fMRI response for the TR $t$, we first find the TR $t^*$ for which the text input yields the highest cosine similarity to the next-token distribution of the text input at TR $t-1$. Next, we isolate the change in fMRI responses following TR $t^*$: we take the difference in the top 100 principal components of the response $R_{t^*} - R_{t^*-1}$ and use them as features. To deal with potential time delays in the fMRI signal, we additionally concatenate these features with the top 100 principal components of $R_{t^*} - R_{t^*-2}$ and $R_{t^*} - R_{t^*-3}$.

In all cases, the induction features are concatenated with the Eng1000 features before being used to linearly predict the fMRI response. When constructing the induction head, we search over the most recent 1024 words and their corresponding fMRI responses. To measure similarity between two texts, we use the predicted next-word distributions yielded by exact ngram matching in the input context ($P_{\text{induction}}^{(\text{exact})}$ in Eq. (5)), which we call *Induction matching*. Alternatively, we can use the predicted next-word distributions yielded by exact ngram matching in the 10B-token OpenWebText reference corpus ($P_{\infty}^{(\text{exact})}$ in Eq. (5)), which we call *Infini-Gram matching*. We additionally explore fuzzy matching techniques in Table A4, but do not see an improvement. This is potentially because the noise and temporal smoothing present in the fMRI response mitigates the benefit of fuzzy matching / matching across fMRI sessions.

**Matching baselines** We add two additional baselines that alter our proposed induction head model only in how they calculate matches. First, *Random matching* selects a random preceding TR as a match. Second, *Naive ngram matching* searches for an exact ngram match in the input context (rather than using the predicted next-word distribution as our induction head does). Specifically, naive ngram matching searches for a match to the most recent 4-word ngram.

### 5.2 INDUCTION MATCHING IMPROVES PREDICTIVE PERFORMANCE

Table 3 shows the fMRI prediction results. Eng1000, the primary interpretable baseline, achieved a mean test correlation of 0.072. In contrast, our model (Induction matching) achieves a mean

---

[6] https://github.com/OpenNeuroDatasets/ds003020

[7] We report results for subject UTS03 due to high fMRI data quality, including superior repeatability, minimal motion, and strong encoding model performance (LeBel et al., 2022).

Table 3: fMRI test prediction performance for different models. Induction matching significantly outperforms other interpretable models. Error bars show 95% CI.

| Feature Model | Mean Correlation | |
|---|---|---|
| | All Voxels | Top 10% Voxels |
| Eng1000 | $0.072 \pm 0.0004$ | $0.220 \pm 0.0012$ |
| Random matching + Eng1000 | $0.069 \pm 0.0003$ | $0.197 \pm 0.0012$ |
| Naive ngram matching + Eng1000 | $0.068 \pm 0.0003$ | $0.194 \pm 0.0012$ |
| Infini-Gram matching + Eng1000 | $0.069 \pm 0.0003$ | $0.200 \pm 0.0012$ |
| **Induction matching** + Eng1000 | $0.087 \pm 0.0005$ | $0.265 \pm 0.0011$ |
| Black-box encodings (LLaMA-2) | $0.096 \pm 0.0005$ | $0.268 \pm 0.0013$ |

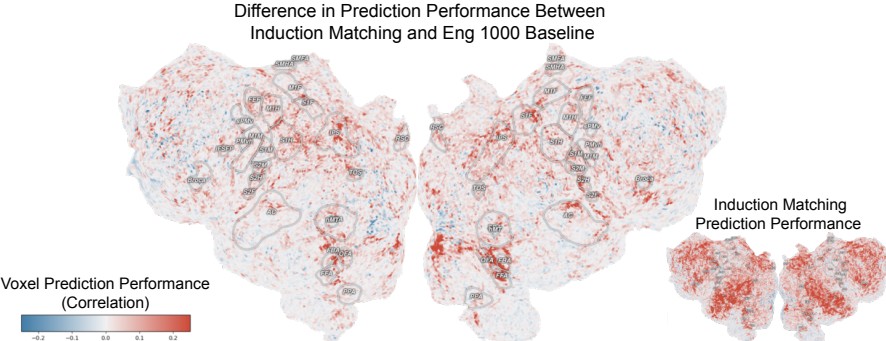

Figure 5: Difference in the correlation performance between the Induction matching and the Eng1000 baseline, visualized across cortex. Performance improvement is scattered across the cortex, but concentrates near some well-studied regions of the language network, *e.g.*, Occipital face area (OFA) and Intraparietal sulcus (IPS).

correlation of 0.087, a 20% improvement over Eng1000. When predicting the top-10% of voxels, Induction Matching achieves a mean correlation of 0.265, again a 20% improvement over Eng1000, and only 1% lower than the black-box LLaMA-2 model (mean correlation 0.268). In contrast, other matching-based baselines are unable to improve over Eng1000. The Naive ngram matching baseline achieves a correlation of 0.068, and the random matching baseline achieves a correlation of 0.069, both of which perform worse than the Eng1000 baseline.

Fig. 5 visualizes the difference in the test correlation performance between the Induction matching and the Eng1000 baseline. The performance improvement (red) is scattered across the cortex, but concentrates near some well-studied regions of the language network, *e.g.*, Occipital face area and Intraparietal sulcus.

**Describing improvements from Induction-Gram** To qualitatively understand the improvements provided by matching, we summarize the text for inputs where different matching procedures (Infini-Gram and Induction) perform well. We use an LLM to do the summarization, following recent works in LLM interpretability (Zhong et al., 2022; Dunlap et al., 2024). We first identify phrases in the input story where a model's performance (average absolute error across voxels) exceeds the baseline performance by more than one standard deviation; see a short example in Fig. 6. Then, we prompt GPT-4 (OpenAI (2023); gpt-4-0613) to generate descriptions for these phrases.

Fig. 6 gives the unedited LLM descriptions[8]. Induction matching is described as capturing *Emotionally or Narratively Critical Phrases*, which aligns with the intuition that Induction improves performance by keeping track of local context in a story, *e.g.*, phrases that "are critical to the plot and character development". In contrast, Infini-Gram matching is described as capturing *Brief, Stand-Alone Phrases*, matching the intuition that Infini-Gram excels in capturing context that is not specific to a particular story, but rather "can stand alone with minimal context". To evaluate the

---

[8]Irrelevant preceding text such as "Sure here is the answer" is removed from the response.

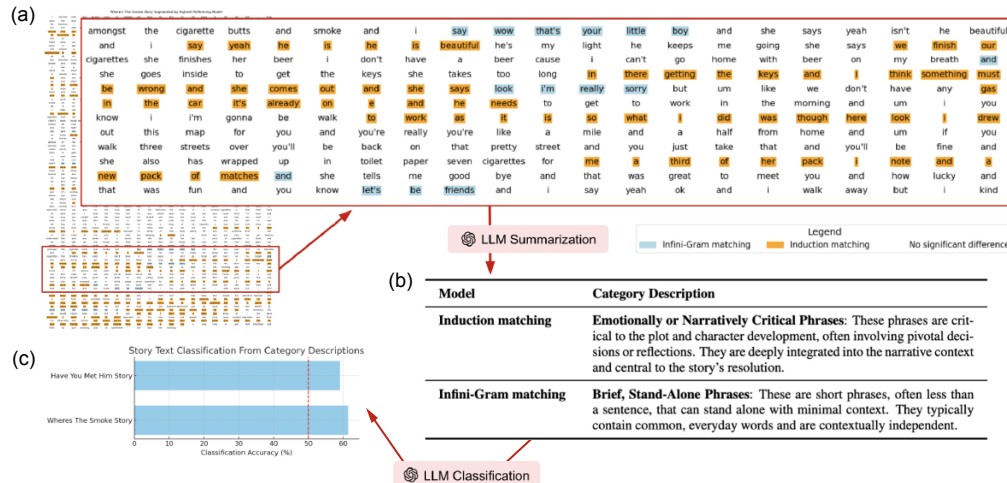

Figure 6: Qualitatively describing where Induction matching / Infini-Gram matching provide improvements. (a) Words in the input story where a model's performance exceeds the baseline performance are highlighted. (b) An LLM summarizes these phrases to yield descriptions for each matching procedure. (c) To check whether these descriptions are faithful, we test whether an LLM can use them to classify the highlighted phrases in the test stories.

accuracy of these descriptions, we prompt GPT-4 to classify the identified phrases in the two test stories using only the descriptions. This yields 61% classification accuracy, a significant (but moderate) improvement over chance (binomial test $p = 0.032$). See all identified phrases and prompts in Appendix A.4.

# 6 DISCUSSION

Induction-Gram constitutes a significant step towards reverse-engineering mechanistically interpretable language models from pre-trained LLMs. Here, we leverage the induction head, which is only one component found to be important in LLMs; future works could integrate new components from mechanistic interpretations, such as indirect object identifiers (Wang et al., 2022), numerical representations (Engels et al., 2024), retrieval heads (Wu et al., 2024), instruction-following heads (Zhang et al., 2023), natural-language explanations of attention heads (Bills et al., 2023) or interpretable submodules within an LLM (Singh et al., 2023b; Bricken et al., 2023). It may be possible to implement these components in a hand-engineered manner, *e.g.*, using python code, regexes, or rule-based models, potentially yielding efficiency in addition to interpretability.

A major limitation of Induction-Gram is that the added induction head provides little improvement when the given input context is short or uninformative. This may be partially mitigated by exploring Induction-Gram in conjunction with techniques such as retrieval-augmented-generation (Wu et al., 2024), that can fetch relevant documents to be incorporated as part of the local context. More generally, while Induction-Gram boasts a very large memory capacity, Induction-Gram relies on ngram-level reasoning and thus continues to struggle with tasks that require significant reasoning capabilities (similar to kNN-LMs (Geng et al., 2024)). Future work may explore the best way to build hybrid models using Induction-Gram and black-box LLMs to achieve effective tradeoffs.

The fMRI analyses conducted here are a suggestive starting point for understanding how context is stored and recalled in the human cortex. Improvements from Induction Matching may help build encoding models that can more rapidly adapt to local context, which can be used in downstream applications such as brain decoding (Tang et al., 2023) or brain-computer interfaces (Nicolas-Alonso & Gomez-Gil, 2012). More generally, the full transparency of Induction-Gram may enable its use in language modeling scenarios that require complete auditing, such as in analyzing scientific text or medical notes (Yang et al., 2023b).

## REPRODUCIBILITY STATEMENT

We include all experimental details necessary for reproduction in the main text and the appendix. For language modeling, explanations of the datasets are provided in Sec. 4.1, and the training details for Fuzzy Matching Model are in Appendix A.1. The inference setup of all models is described in Appendix A.3. For the natural-language fMRI experiment, details about the constructing induction-based input features are described in Sec. 5.1. Details about the publicly available data set, data collection methods, and the procedures used to map embedded stimuli to BOLD responses are provided in Appendix A.4.

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

## A APPENDIX

### A.1 TRAINING OF FUZZY MATCHING MODEL

**Architecture of Fuzzy Matching Model** We train two Fuzzy Matching Models, one using the GPT-2 tokenizer and the other using the LLaMA-2 tokenizer. With GPT-2 tokenizer, Fuzzy Matching Model consists of four transformer layers, whereas it comprises three transformer layers when using LLaMA-2 tokenizer. Since relative position is crucial for calculating similarity, we incorporate Relative Positional Encoding (Shaw et al., 2018), with a maximum relative position of 32 for the GPT-2 tokenizer and 64 for the LLaMA-2 tokenizer. The vocabulary embeddings are initialized with those from GPT-2 and LLaMA2-7B, ensuring that the number of heads and embedding dimensions align with the specifications of GPT-2 and LLaMA2-7B.

**Creating Similarity pair with LLMs** For both Fuzzy Matching Model, we use LLaMA2-7B as a teacher model. OpenWebText and Pile-train[9] datasets for training each Fuzzy Matching Model thats use GPT-2 or LLaMA-2 tokenizer. During training, we randomly sample sequences of 32 or 64 tokens with batch size of 128 or 256, resulting in 4,096 or 16,384 next-token prediction probabilities per batch. From these, we sample distant 3,584 or 4,096 queries and 512 keys and create similarity pairs ($3,584 \times 512$ or $4,096 \times 512$) by calculating similarity based on Equation (5). The models are trained using a combination of CE loss and reverse KLD loss, with equal weights (1.0). We adopt most of the training settings from the codebase[10] for training. Gradients are accumulated over 16 iterations, and we use the AdamW optimizer (Loshchilov & Hutter, 2019) with a learning rate of 0.0001 and a weight decay of 0.1. The learning rate follows a cosine schedule with a warmup

Table A1: Ablation study on training of Fuzzy Matching Model. Next-token accuracy (%) of Induction-only (fuzzy) on the BabyLM-test is reported. LLaMA-2 tokenizer is used.

| Positional Encoding | Reverse KLD loss | Forward KLD loss | CE loss | Accuracy |
|---|---|---|---|---|
| Relative | ✓ | | ✓ | 43.2 |
| Relative | | ✓ | ✓ | 42.8 |
| Relative | | | ✓ | 42.7 |
| Relative | ✓ | | | 41.9 |
| Sinusoidal | ✓ | | ✓ | 37.0 |

over the first 1,000 iterations, and training continues for 15,000 or 20,000 iterations. Training is conducted on four NVIDIA A100 GPUs. **NEW**

**Ablation Study on Fuzzy Matching Model Training**   We conduct an ablation study on the positional encoding strategy and training process of Fuzzy Matching Model using the OpenWebText dataset to distill it from LLaMA-2-7B. The study evaluates the contributions of Relative Positional Encoding, reverse KLD loss, and CE loss to the model's effectiveness. As shown in Table A1, next-token prediction accuracy improves significantly when both reverse KLD and CE losses are included, demonstrating their complementary roles in optimizing the Fuzzy Matching Model. With CE loss, Forward KLD loss is less effective than reverse KLD loss. Furthermore, using Relative Positional Encoding instead of Sinusoidal Positional Encoding leads to better performance, highlighting the advantages of incorporating relative positional information for enhanced fuzzy matching capabilities.

## A.2   DETERMINATION OF $\tau$

To build Induction-Gram by integrating the three types of estimations, we first need to determine the threshold for effective $n$, denoted as $\tau$. To identify the optimal value of $\tau$, we conducted cross-validation using the BabyLM training set (100M tokens). BabyLM consists of six datasets: open_subtitles, bnc_spoken, gutenberg, childes, simple_wiki, and switchboard. Since switchboard contains only 2M tokens, we exclude it from the experiment. For the remaining datasets, we use each dataset as a validation set, while the other four are used as the reference corpus to build Infini-Gram. We then compare the performance changes of Infini-Gram, Induction-only (exact), and Induction-only (fuzzy) depending on effective $n$. 10k samples are used for evaluating on each dataset.

As shown in Figure A1, Infini-Gram outperforms Induction-only (exact) when the effective $n$ exceeds 8 for the GPT-2 tokenizer and 9 for the LLaMA-2 tokenizer. Therefore, we set $\tau$ to 8 and 9 for the respective tokenizers.

## A.3   LANGUAGE MODELING RESULTS EXTENDED

**Experimental Details**   We use diverse datasets as reference corpus for Infini-Gram. We use Infini-Gram that is released by authors[11] for Pile-train[12] and Pile-val[13]. For BabyLM-dev and OpenWebText, we build our own Infini-Gram. We use public code to build and inference Infini-Gram[14] and Induction-only (exact)[15]. During inference, the maximum length for exact matching with Infini-Gram is 500, and we use window size $k$ for fuzzy matching as 32 and 64 for GPT-2 and LLaMA-2 tokenizers, respectively. **NEW**

---

[9] https://huggingface.co/datasets/monology/pile-uncopyrighted
[10] https://github.com/karpathy/minGPT
[11] https://infini-gram.io/api_doc.html
[12] v4_piletrain_llama
[13] v4_pileval_llama and v4_piletrain_gpt2
[14] https://infini-gram.io/pkg_doc.html
[15] https://github.com/AlexWan0/infini-gram/tree/main

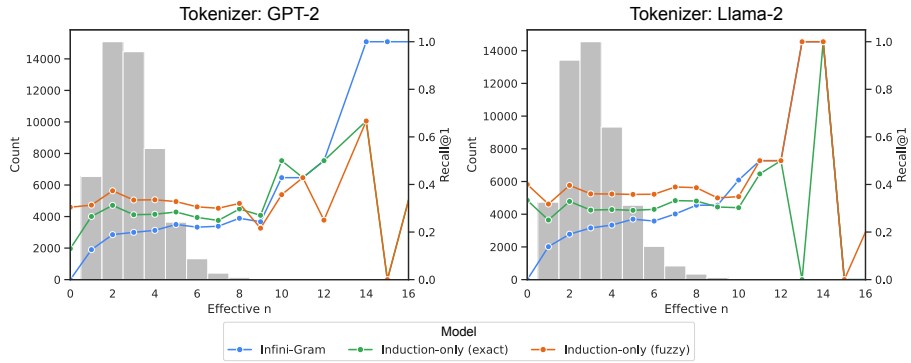

Figure A1: Comparison of next-token accuracy.

Table A2: Ablation study on components of **Induction-Gram**. Next-token accuracy (%) on BabyLM-test is reported.

| Reference Corpus | BabyLM-dev | Pile-val | OpenWebText | Pile-train |
|---|---|---|---|---|
| Induction-Gram | 43.1 | 42.9 | 43.2 | 49.4 |
| w/o Induction-only (fuzzy) | 42.2 | 36.9 | 38.3 | 46.6 |
| w/o Induction-only (exact) | 43.0 | 42.8 | 43.1 | 49.3 |
| w/o Infini-Gram | | 42.9 | | |
| Infini-Gram | 39.0 | 19.0 | 20.1 | 33.5 |

**Ablation Study on Induction-Gram** We conduct an ablation study to assess the impact of each component in Induction-Gram. Table A2 reports next-token accuracy when individual components are omitted. Excluding Induction-only (fuzzy) results in a more significant performance drop than removing Induction-only (exact). This underscores the importance of fuzzy matching in handling diverse contexts and improving adaptability, as reflected in Table 1, where Induction-only (fuzzy) outperforms Induction-only (exact). Since both components act as induction heads, they exhibit complementary roles—when one is removed, the other partially compensates for its absence. Only when using Pile-train as a reference corpus, omitting Infini-Gram leads to the most substantial performance decline. It is worth noting that when the reference corpus lacks similarity to the test dataset's distribution (*e.g.*, Pile-val, OpenWebText, and Pile-train), the performance of Infini-Gram falls significantly below the scenario where it is not utilized at all. This highlights the sensitivity of Infini-Gram to the quality and relevance of the reference corpus.

**Speculative Decoding Results Extended** Table A3 reports the inference times for Induction-only (fuzzy) and Induction-Gram using speculative decoding, with the OpenWebText dataset serving as the reference corpus for Infini-Gram. We find matches with a maximum of 64 tokens for both exact and fuzzy matching. The experiments are conducted on two NVIDIA H100 GPUs and 64 CPU cores. Although Induction-Gram requires more time for generation on average than Induction-only (fuzzy), it still significantly reduces inference time compared to relying solely on a large model for inference.                                                                                  NEW

**Explanation** Figure A2 presents several examples of explanations provided by Induction-Gram. Even if an exact match fails to yield a good match, when the probability of subsequent tokens is similar, the fuzzy matching model can predict with high similarity, enabling successful fuzzy matching, enabling successful fuzzy matching, and improving next-token prediction.

**Exact Matching within Context**

**(a) Input Prompt:** "... `_Frontispiece_--(_Page 61_)]` `\nBUNNY BROWN AND HIS`                              **"ER"**

| Sequence from Context | Effective n | Next Token |
|---|---|---|
| "... `PG70358 = = =` `\nBUNNY BROWN AND HIS SIST`" | 13 | ER |

**(b) Input Prompt:** "... `Then the chorus:` `"Will you, won't you, will you, won'`"                              **"t"**

| Sequence from Context | Effective n | Next Token |
|---|---|---|
| "... `out in a friendly voice:\n`"Will you, won't you, will you, won'`"" | 13 | t |

**(c) Input Prompt:** "... `Breuschwickersheim` `is a commune. It is in Grand Est in the`"                              **"Bas"**

| Sequence from Context | Effective n | Next Token |
|---|---|---|
| "... `Elsenheim` `is a commune. It is in Grand Est in the`" |  | Bas |
| "... `Ohnenheim` `is a commune. It is in Grand Est in the`" | 12 | Bas |
| "... `Bourgheim` `is a commune. It is in Grand Est in the`" |  | Bas |

**Fuzzy Matching within Context**

**(d) Input Prompt:** "... `Simpson still delays taking the kick, now it comes`"                              **"in"**

| Sequence from Context | Similarity | Next Token |
|---|---|---|
| "... `a great breakaway down the left, the cross coming`" | 0.160 | in |
| "... `Three minutes later Simpson ran`" | 0.083 | on |
| "... `but he grabbed it again at the second attempt before it went`" | 0.075 | over |
| "... `but he was forced just a little bit wide. \nHe ran`" | 0.075 | into |
| "... `to blow it for half time, United skipper,  Steve Foster drove`" | 0.072 | forward |

**(e) Input Prompt:** "... `Because he says it's Lincolnshire ! \nNo, he didn't! \nHe said`"                              **"it"**

| Sequence from Context | Similarity | Next Token |
|---|---|---|
| "... `What's Lincolnshire gotta do with it? \nBecause he says`" | 0.680 | it |
| "... `God that wind's gone cold! \nI say`" | 0.210 | that |
| "... `Well he don't know anything about gardening, you see! \nBut`" | 0.203 | I |
| "... `What's Lincolnshire gotta do with it? \nBecause`" | 0.186 | he |
| "... `I don't know why`" | 0.179 | ! |

**(f) Input Prompt:** "... `So I taught him that the first week, and the second`"                              **"week"**

| Sequence from Context | Similarity | Next Token |
|---|---|---|
| "... `And I was running it and the first`" | 0.098 | week |
| "... `who's erm sixty odd and he comes in here every`" | 0.087 | day |
| "... `And I was running it and the first week I got there, and one`" | 0.053 | gu |
| "... `So I taught him that the first`" | 0.042 | week |
| "... `we had to cancel because nobody turned up.\nEr one`" | 0.035 | of |

Figure A2: Examples of explanation of Induction-Gram from BabyLM-test. (a), (b), and (c) show examples of exact matching while (d), (e), and (f) show examples of fuzzy matching.

Table A3: Speed of speculative decoding (SP). The mean and standard deviation of 3 runs are reported.

| Draft Model | Large Model | SP | BabyLM-test | | Pile-val | | FineWeb | |
|---|---|---|---|---|---|---|---|---|
| | | | ms/token ($\downarrow$) | Speed Up ($\uparrow$) | ms/token ($\downarrow$) | Speed Up ($\uparrow$) | ms/token ($\downarrow$) | Speed Up ($\uparrow$) |
| | LLaMA2-13B | | 26.4±0.1 | | 26.3±0.4 | | | |
| Induction-only (fuzzy) | LLaMA2-13B | ✓ | 13.3±0.2 | 1.98 | 14.9±0.1 | 1.77 | 14.9±0.3 | 1.76 |
| Induction-Gram | LLaMA2-13B | ✓ | 23.1±0.4 | 1.14 | 22.8±0.3 | 1.15 | 23.0±0.7 | 1.14 |
| | LLaMA2-70B | | 71.2±0.1 | | 71.0±0.2 | | 71.1±0.2 | |
| Induction-only (fuzzy) | LLaMA2-70B | ✓ | 31.4±0.7 | 2.27 | 33.3±0.6 | 2.13 | 33.2±1.0 | 2.15 |
| Induction-Gram | LLaMA2-70B | ✓ | 42.0±0.7 | 1.70 | 41.6±1.0 | 1.71 | 40.4±1.2 | 1.76 |

## A.4 FMRI RESULTS EXTENDED

**Data details**   This section gives more details on the fMRI experiment we analyze. These MRI data are available publicly (LeBel et al., 2022; Tang et al., 2023), but the methods are summarized here. Functional magnetic resonance imaging (fMRI) data were collected from 3 human subjects as they listened to English language podcast stories over Sensimetrics S14 headphones. Subjects were not asked to make any responses, but simply to listen attentively to the stories. For encoding model training, each subject listened to at approximately 20 hours of unique stories across 20 scanning sessions, yielding a total of ~33,000 datapoints for each voxel across the whole brain. For model testing, the subjects listened to two test stories 5 times each, and one test story 10 times, at a rate of 1 test story per session. These test responses were averaged across repetitions. Functional signal-to-noise ratios in each voxel were computed using the mean-explainable variance method from (Nishimoto et al., 2017) on the repeated test data. Only voxels within 8 mm of the mid-cortical surface were analyzed, yielding roughly 90,000 voxels per subject.

MRI data were collected on a 3T Siemens Skyra scanner at University of Texas at Austin using a 64-channel Siemens volume coil. Functional scans were collected using a gradient echo EPI sequence with repetition time (TR) = 2.00 s, echo time (TE) = 30.8 ms, flip angle = 71°, multi-band factor (simultaneous multi-slice) = 2, voxel size = 2.6mm x 2.6mm x 2.6mm (slice thickness = 2.6mm), matrix size = 84x84, and field of view = 220 mm. Anatomical data were collected using a T1-weighted multi-echo MP-RAGE sequence with voxel size = 1mm x 1mm x 1mm following the Freesurfer morphometry protocol (Fischl, 2012).

All subjects were healthy and had normal hearing. The experimental protocol was approved by the Institutional Review Board at the University of Texas at Austin. Written informed consent was obtained from all subjects.

All functional data were motion corrected using the FMRIB Linear Image Registration Tool (FLIRT) from FSL 5.0. FLIRT was used to align all data to a template that was made from the average across the first functional run in the first story session for each subject. These automatic alignments were manually checked for accuracy.

Low frequency voxel response drift was identified using a 2nd order Savitzky-Golay filter with a 120 second window and then subtracted from the signal. To avoid onset artifacts and poor detrending performance near each end of the scan, responses were trimmed by removing 20 seconds (10 volumes) at the beginning and end of each scan, which removed the 10-second silent period and the first and last 10 seconds of each story. The mean response for each voxel was subtracted and the remaining response was scaled to have unit variance.

We used the fMRI data to generate a voxelwise brain encoding model for natural language using different encoding models. In order to temporally align word times with TR times, Lanczos interpolation was applied with a window size of 3. The hemodyanmic response function was approximated with a finite impulse response model using 4 delays at -8,-6,-4 and -2 seconds (Huth et al., 2016). For each subject $x$, voxel $v$, we fit a separate encoding model $g_{(x,v)}$ to predict the BOLD response $\hat{B}$ from our embedded stimulus, i.e. $\hat{B}_{(x,v)} = g_{(x,v)}(H_i(\mathcal{S}))$. To evaluate the voxelwise encoding models, we used the learned $g_{(x,v)}$ to generate and evaluate predictions on a held-out test set.

Table A4: fMRI Prediction Performance when using fuzzy matching. Error bars show 95% CI.

| Feature Model | Tokenizer | Matching Model | Mean Correlation | |
|---|---|---|---|---|
| | | | All Voxels | Top 10% Voxels |
| Eng1000 | - | - | $0.072 \pm 0.0004$ | $0.220 \pm 0.0012$ |
| Infini-Gram + Eng1000 | GPT-2 | - | $0.069 \pm 0.0003$ | $0.200 \pm 0.0012$ |
| Induction Matching + Eng1000 | GPT-2 | - | $0.087 \pm 0.0005$ | $0.265 \pm 0.0011$ |
| Fuzzy Induction Matching + Eng1000 | GPT-2 | GPT-2 | $0.076 \pm 0.0004$ | $0.222 \pm 0.0011$ |
| Fuzzy Induction Matching + Eng1000 | LLaMA-2 | LLaMA2-70B | $0.076 \pm 0.0004$ | $0.225 \pm 0.0012$ |
| Fuzzy Induction Matching + Eng1000 | GPT-2 | Fuzzy Matching Model | $0.076 \pm 0.0004$ | $0.216 \pm 0.0011$ |
| Fuzzy Induction Matching + Eng1000 | LLaMA-2 | Fuzzy Matching Model | $0.077 \pm 0.0004$ | $0.223 \pm 0.0012$ |

**fMRI fuzzy induction head settings**     Similar to the Exact Induction Matching technique described in Sec. 5.1, we construct an induction head for fuzzy matching. In the fuzzy setting, we leverage the predicted next-word distributions obtained through fuzzy n-gram matching in the input context ($P_{\text{induction}}^{(\text{fuzzy})}$ in Equation (3)), which we refer to as *Fuzzy Induction Matching*. Specifically, we calculate the cosine similarity between the next-word distributions of the current word and all prior candidate words.

To account for the temporal resolution of fMRI, we apply Lanczos smoothing to the word-level similarity values, aligning these values with the fMRI time scale. This allows us to identify the time point (TR) $t^*$ that maximally corresponds to the current time point $t$ based on the highest similarity.

We evaluate several configurations for deriving the next-word distributions, including GPT-2, LLaMa-2, the Fuzzy Matching model with the GPT-2 tokenizer, and the Fuzzy Matching Model with the LLaMA-2 tokenizer. See more details on Fuzzy Matching models in Sec. 3.2.

**Extended prediction performance results**     The prediction performance of Fuzzy Induction Matching Models is compared to the performance of the Exact Induction Matching Models and the Eng1000 baseline in Table A4. The Fuzzy Induction Model, in its highest-performing configuration (using the Fuzzy Matching Model with the LLaMa2-70B tokenizer), achieves only a 6.94% improvement in prediction performance compared to the Eng1000 baseline. The lower relative performance of Fuzzy Induction Matching compared to Exact Induction Matching may be due to the inherent noise and lower spatial and temporal resolution of fMRI data, which makes it challenging to detect subtle differences in neural activations associated with similar but non-identical stimuli.

| Title | Prompt |
|---|---|
| GPT-4 Prompt for Generating Category Descriptions | I have provided two test stories below. Specific phrases from each story have been picked out based on the performance of different encoding models. Can you describe the characteristics of the words and phrases that each category contains? Be specific about the type of words, their context in the story, and any other relevant commonalities. Write succinct descriptions for each category that would allow one to categorize phrases in other such stories accurately.
Category A: ['sh first she digs into her cutoffs in the', 'both need this right now i', ... ]
Category B: ['to everything or you make yourself scarce', 'my cigarettes and uh', ...]
Full Story: [['i reached over and secretly'], ['undid my seatbelt'], ...] |
| GPT-4 Prompt for Classifying Stages Based on Descriptions | I have attached category descriptions below. Based on the descriptions, in order, go through each short list of words (short phrase) in the story at the end and classify the segments into one of the categories. Rather than listing all the phrases in a category at a time, list each phrase in order and label it as belonging to category A or B.
Category A: Emotionally, or Narratively Critical ...
Category B: Brief, Stand-Alone Phrases ...
Full Story: [['i reached over and secretly'], ['undid my seatbelt'], ...] |

Table A5: GPT-4 Prompts for Generating and Classifying Categories of Text. Ellipses (...) indicate omitted portions of the full prompts.

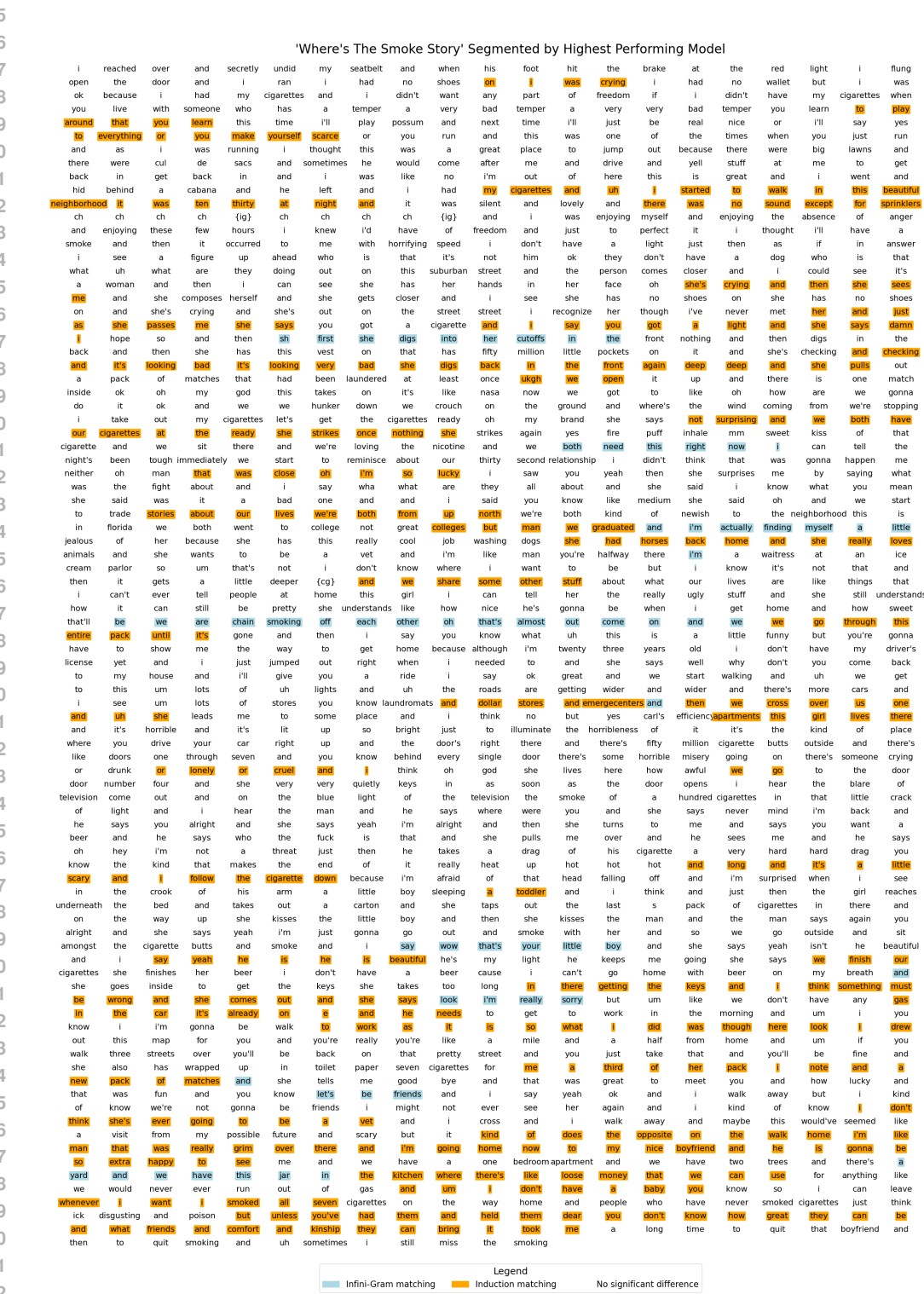

Figure A3: Test story 1 (*Where's There's Smoke*), highlighted in regions where the Infini-Gram matching and Induction matching models exceed baseline performance, measured by the average absolute error across voxels, by more than one standard deviation.

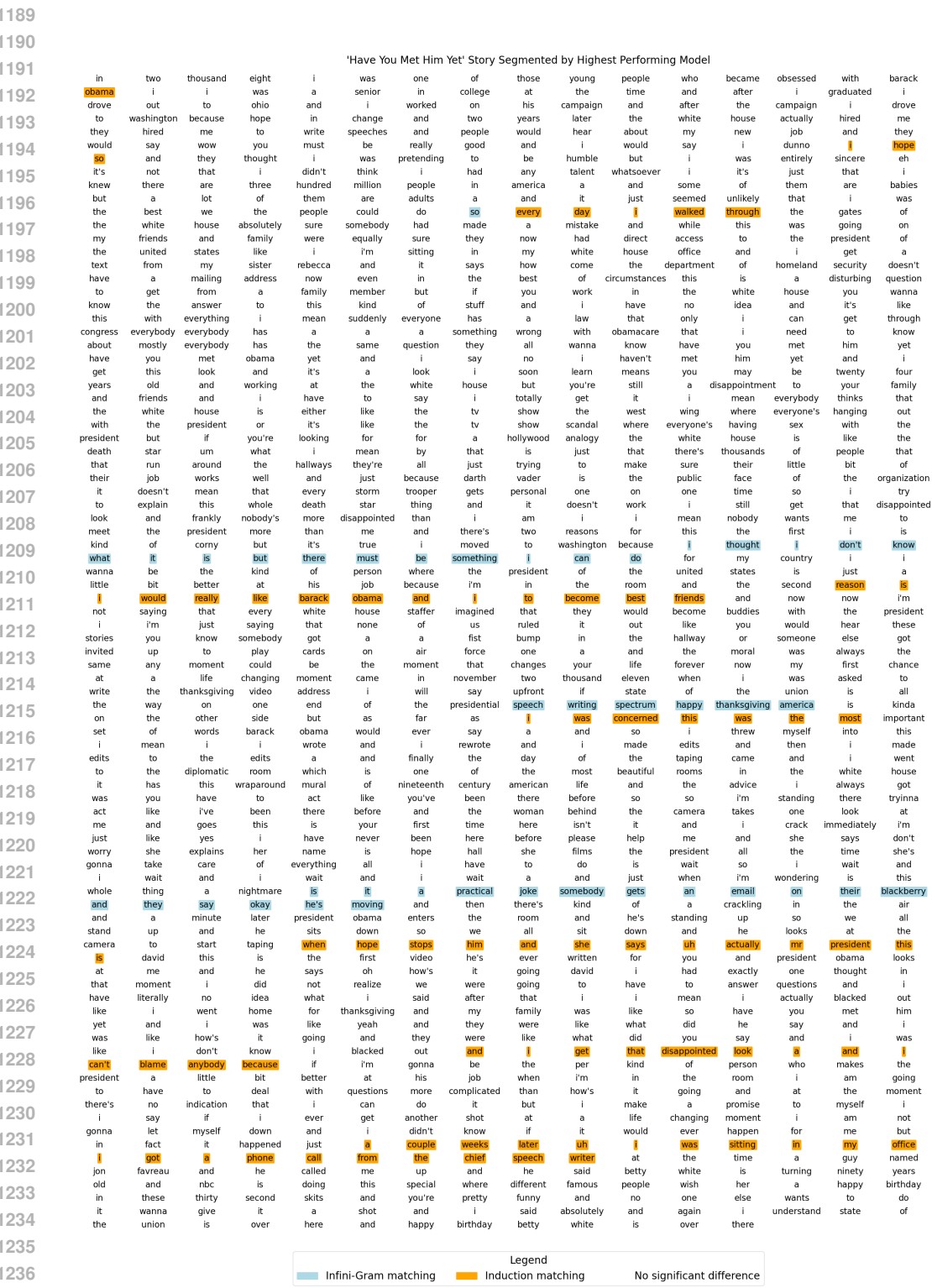

Figure A4: The first section of test story 2 (*Have You Met Him Yet*), highlighted in regions where the Infini-Gram and Induction matching models exceed baseline performance.

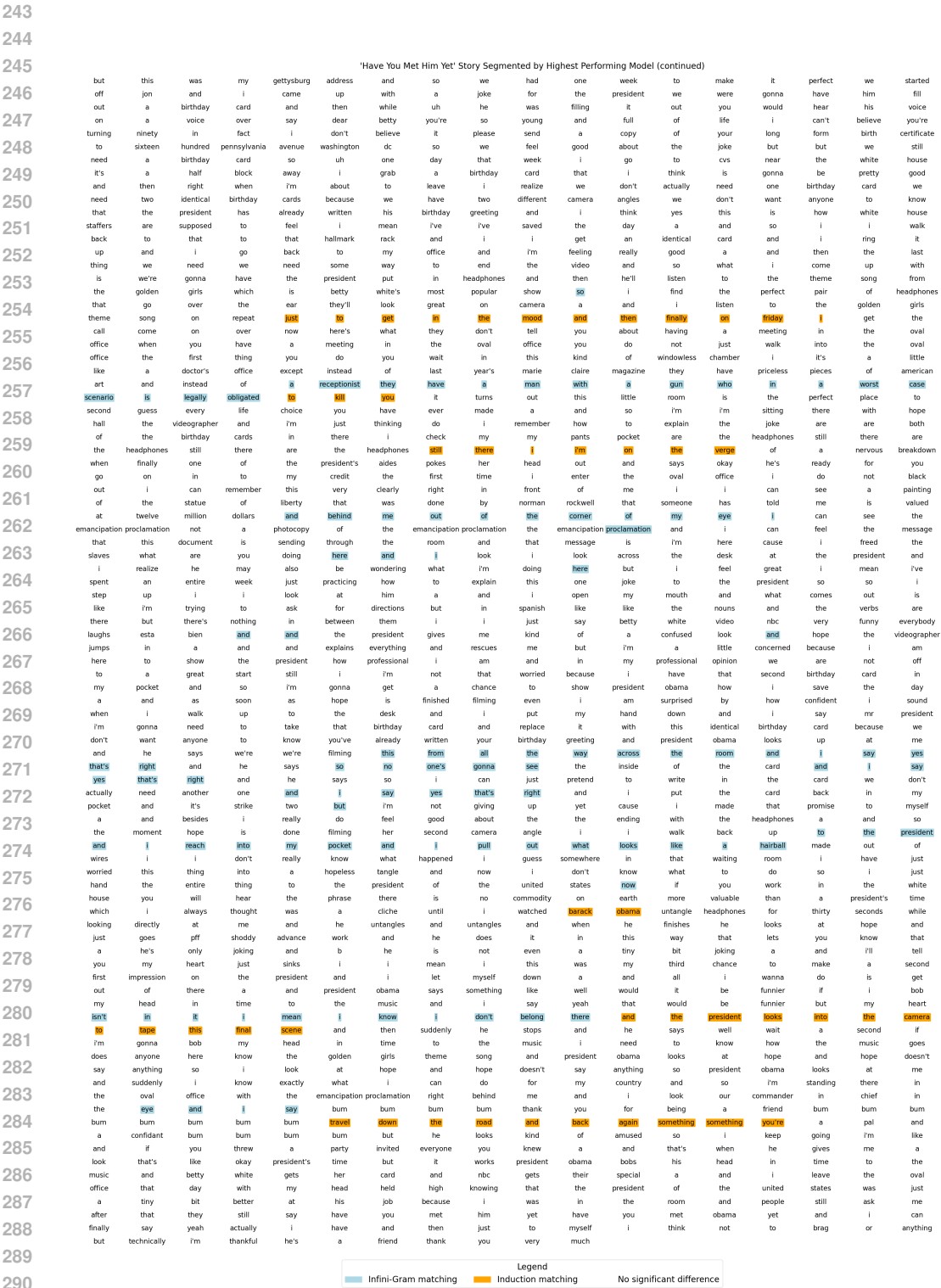

Figure A5: The second section of test story 2 (*Have You Met Him Yet*), highlighted in regions where the Infini-Gram and Induction matching models exceed baseline performance.

