# OpenReview forum: "Interpretable Language Modeling via Induction-head Ngram Models"
_ICLR.cc/2025/Conference — Submitted to ICLR 2025_

### Official Review · Reviewer_RdyW · 2024-10-29

**Soundness:** 4
**Presentation:** 3
**Contribution:** 3
**Rating:** 8
**Confidence:** 3

**Summary:**

This paper proposes a new interpretable-by-design approach for language modeling and similar prediction tasks named Induction-Gram, in which the existing Infini-gram approach by [Liu et al. (2024)](https://openreview.net/forum?id=u2vAyMeLMm#discussion) for n-gram language modeling is combined with an handcrafted induction mechanism inspired by the induction heads observed to emerge in large language models during early training stages. The proposed approach is shown to improve next word prediction performance for shorter n-gram sequences where the Infini-gram approach underperforms by using local context. Moreover, authors experiment with using their proposed fuzzy induction approach as a draft model for speculative decoding, accelerating LLM generation in a similar or better way than current approaches. Finally, the proposed induction methodology is adapted to predict brain responses to natural language stimuli recorded via fMRI. Again, the induction method outperforms simpler matching baselines, obtaining performances short of state-of-the-art black-box LLM embeddings. Finally, authors employ LLMs to identify common properties of texts for which various prediction strategies perform well, finding induction matching to perform well on narratively critical phrases relying on preceding context.

**Strengths:**

The idea of complementing n-gram modeling with interpretable context-based mechanisms proposed in this work is very appealing since an interpretable-by-design model mirroring the in-weights vs. in-context learning of modern LLMs can be used to further our understanding of how these mechanisms operate and when they fail. The proposed combination of fuzzy induction matching and Infini-gram is simple yet elegant and seems to address the shortcomings of n-gram-based approaches in an effective way. The fMRI analysis provides an interesting perspective on extending such a technique to other sequence prediction tasks, showcasing promising results. Finally, the qualitative evaluation of Induction-gram improvements further supports the relevance of the proposed approach in context-rich settings.

**Weaknesses:**

According to Table 1, Induction (fuzzy) would be a strong contender for the best method across all tested settings when using the GPT-2 tokenizer, but falls significantly behind Induction-gram for the LLaMA-2 tokenizer. While this was not discussed in depth in the paper, I believe this results could indicate ta lack of robustness of the reported findings across various tokenization systems.

**Minor corrections:**

Line 238: Typo "are is low"

Mentions to BabyLM should cite the work [Findings of the BabyLM Challenge: Sample-Efficient Pretraining on Developmentally Plausible Corpora
](https://aclanthology.org/2023.conll-babylm.1/) corresponding to the first edition of the shared task.

**Questions:**

Is the choice of a Fuzzy Matching Model constrained to models using the same tokenizer as the Infini-gram when combining them for Induction-gram? Otherwise, the choice of retraining a similarity model rather than using pretrained sentence embeddings for fuzzy retrieval is unclear to me.

In Figures 2b and 4, it is shown that the large majority of ngram predictions on BabyLM use an effective n below ~8, even when using in-distribution data from another BabyLM split to create the ngram model. In light of this, the choice of the $\tau = 8$ or $9$ in Equation 5 seems to suggest that most cases will recur to fuzzy matching. In light of this, is it even meaningful to maintain the n-gram component in Induction-gram, or are trends shown in these Figure not representative of other tested datasets?

---

> ### Author Response · Authors · 2024-11-22
>
> Thank you for your thoughtful and positive feedback on our work. Your recognition of the simplicity and elegance of our proposed methods is truly encouraging. Below, we address your concern in more detail.
>
> **W1: Robustness across Tokenization Systems**
>
> **1) Performance Variation Across Tokenizers**
>
> While the numerical performance of Induction-Gram differs between the GPT-2 and LLaMA-2 tokenizers, our analysis in Table 1 shows that the relative improvement from Infini-Gram to Induction-Gram is consistent across both systems. This suggests that the core mechanism of Induction-Gram operates reliably regardless of the tokenizer, even though absolute performance metrics may vary.
>
> **2) Factors Affecting Absolute Performance**
>
> The observed performance disparity can be explained by two key characteristics of the LLaMA-2 tokenizer:
> Increased Reference Tokens: The LLaMA-2 tokenizer generates more granular tokens, providing a richer set of references for the induction mechanism to utilize effectively.
> Reduced Prediction Space: With a smaller vocabulary size (32k vs. GPT-2's 50k), the LLaMA-2 tokenizer simplifies the next-token prediction task, which inherently improves accuracy.
>
> These differences highlight the interaction between tokenization systems and model performance but do not diminish the generality or robustness of Induction-Gram’s core mechanisms.
>
> **Minor Corrections**
>
> Additionally, thank you for pointing out the minor corrections. We have incorporated these into the revised version of the paper, with the changes highlighted in blue.

---

### Official Review · Reviewer_YaGh · 2024-11-02

**Soundness:** 2
**Presentation:** 2
**Contribution:** 3
**Rating:** 5
**Confidence:** 4

**Summary:**

The paper presents a method called the Induction-head ngram model (Induction-Gram), aimed at addressing the issues of interpretability and efficiency in large language models (LLMs). By combining modern ngram models with a hand-designed "induction head," Induction-Gram improves interpretability while significantly enhancing the accuracy of next-word prediction, demonstrating good performance in the natural language fMRI context. Overall, the paper has a high level of innovation and research value, but there are also several areas that require improvement.

**Strengths:**

The paper demonstrates significant methodological and application innovations with the Induction-Gram model. By introducing the induction head into the ngram framework, the model effectively addresses the limitations of traditional ngram models in adapting to new contexts and handling ambiguous matches. This approach offers a fresh perspective for building interpretable and efficient language models, showcasing notable originality compared to existing research. Furthermore, applying the Induction-Gram model in natural language fMRI contexts is a pioneering effort that broadens the application scope of language models and provides new tools and insights for neuroscience research. The study contributes to a deeper understanding of the mechanisms underlying language models, particularly the role of the induction head in contextual learning. It provides a theoretical basis for advancing the research on the interpretability of language models. Practically, the model enhances prediction accuracy while maintaining interpretability, which is crucial for high-stakes applications requiring explainable model decisions, such as in science, medicine, and policy-making.

**Weaknesses:**

The limitations in comparison with other models: The paper only compares the Induction-Gram model with a few baseline models, making it difficult to comprehensively assess its relative strengths and weaknesses in the broader modeling field. The lack of comparisons with more diverse types of models, particularly those with unique advantages in interpretability and efficiency, fails to clearly demonstrate the model's true competitiveness in different modeling environments and hinders the identification of its optimal applicability across various scenarios. Insufficient explanation of experimental parameter settings: For some key experimental parameters, such as the hyperparameters in the Fuzzy Matching Model, the paper does not provide sufficient detail regarding their selection criteria and analysis of their impact on model performance. This makes it challenging for other researchers to understand the rationale behind these parameter settings, hindering their ability to effectively adjust and optimize parameters according to their research needs, which ultimately affects the model's reproducibility and scalability.

**Questions:**

1.Regarding the potential improvements of the induction head: The authors mention that under suboptimal input contexts, techniques such as retrieval-augmented generation can be combined to enhance the performance of the induction head. Can they elaborate on the specific mechanisms of this combination and the expected outcomes?
2.Details of comparison with other models: The paper only compares the Induction-Gram model with a limited number of baseline models. To provide a more comprehensive assessment of the model's performance, it would be helpful to understand its specific advantages and disadvantages compared to other models that excel in interpretability and efficiency. Particularly, how does the Induction-Gram model's performance differ from these models when handling complex linguistic structures, domain-specific texts, and large-scale data?
3.Impact of training data on the Fuzzy Matching Model: The paper states that the Fuzzy Matching Model was trained using the OpenWebText and Pile-train datasets but does not elaborate on how the selection of these datasets specifically impacts model performance. Different datasets may exhibit varying language styles, topic distributions, and data quality, which could affect the similarity metrics learned by the Fuzzy Matching Model. How do these factors influence the performance of the Induction-Gram model across different tasks?

---

> ### Author Response · Authors · 2024-11-22
>
> Thank you for your thoughtful and encouraging feedback on our work. We deeply appreciate your recognition of the methodological contributions and practical implications of the Induction-Gram. Below, we provide additional clarifications and present experimental results to further address your points and concerns. Some changes have also been incorporated into the revised manuscript, highlighted in blue for your convenience.
>
> **W1 (Q2): Comparison with Other models**
>
> First, we would like to clarify our rationale for the selected baselines and the scope of our comparisons:
>
> As outlined in the paper:
> > "... there is a considerable gap between the performance of interpretable models and black-box LLMs in next- token prediction. There is a considerable gap between the performance of interpretable models and black-box LLMs in next-token prediction."
>
> Our primary goal in this work is to develop an inherently interpretable model that offers competitive performance. In the current landscape of language modeling, interpretability is often addressed through post-hoc methods that aim to analyze or explain black-box models after training. While these approaches can provide insights into model behavior, they do not inherently build transparency into the prediction process, limiting their ability to offer meaningful rationales.
>
> Given this context, few models are designed to prioritize both interpretability and performance. We chose Infini-Gram as our primary baseline because it is a state-of-the-art interpretable ngram model and directly aligns with our focus on inherently interpretable modeling. Comparisons with black-box models (e.g., GPT-2, LLaMA) were included to illustrate the performance gap between interpretable and black-box models. Models focusing solely on efficiency or black-box optimization techniques (e.g., pruning, distillation) were excluded, as they do not align with our primary objective of interpretability.
>
> **Relative Advantages and Disadvantages of Induction-Gram**
>
> The key advantage of Induction-Gram lies in its inherent interpretability. Unlike black-box LLMs, Induction-Gram provides transparent predictions by leveraging ngram properties, ensuring that each predicted token is grounded in a data-driven, explainable process. This approach adheres to the principles of interpretable modeling and enhances trust in the model’s outputs.
>
> In addition, Induction-Gram addresses major limitations of Infini-Gram, as noted in our paper:
> > "Infini-Gram struggles with adapting to new contexts and with matching queries that cannot be found exactly within a reference dataset (e.g., typos or rephrasings)."
>
> To overcome these challenges, we introduced induction-head mechanisms and fuzzy matching, which improve adaptability and robustness without compromising interpretability. As shown in Table 1, these enhancements lead to significant performance improvements over Infini-Gram, particularly in handling complex linguistic patterns.
>
> While Induction-Gram introduces clear advantages, it also comes with trade-offs. The primary cost is increased resource usage, as building and running the induction mechanism requires additional computational resources. We acknowledge this limitation and view it as an area for future optimization.
>
> **W2: Ablation Study on Training the Fuzzy Matching Model**
>
> We included details of Fuzzy Matching Model in Section A.1 of the Appendix. Additionally, we conducted an ablation study to evaluate the effects of key components, including transformer block types (Relative Position Encoding, RPE vs. Sinusoidal) and the two loss terms (reverse KLD and CE).
> The results, summarized in the table below, illustrate the impact of these components on the performance of the Induction-Only (fuzzy) model on the BabyLM-test dataset (using LLaMA-2 tokenizer). Here, Fuzzy Matching Models were trained on the OpenWebText dataset.
>
> | Positional Encoding | Reverse KLD loss | Forward KLD loss | CE loss | Performance |
> |---|---|---|---|---|
> | RPE | O |  | O | 43.2 (Ours)|
> | RPE |  | O | O | 42.8 |
> | RPE |  |  | O | 42.7 |
> | RPE | O |  |  | 41.9 |
> | Sinusoidal | O |  | O | 37.0 |
>
> Key observations are:
> * Combining reverse KLD loss and CE loss leads to significant performance improvements, demonstrating their complementary roles in optimizing the Fuzzy Matching Model.
> * Reverse KLD is more effective than forward KLD, suggesting its suitability for this context.
> * Relative Position Encoding (RPE) outperforms the Sinusoidal transformer block, likely due to its ability to better capture contextual relationships and positional information.
>
> This ablation study and detailed analysis have been included in the Appendix.

---

> ### Author Response · Authors · 2024-11-22
>
> **Q1: Potential Integrations with Other Methods**
>
> There are many interesting potential directions involving bridging Induction-Gram with more powerful methods to improve performance in difficult input contexts. For example, Induction-Gram can struggle with very short or under-specified contexts (e.g., a single-sentence question), as the input does not provide enough contextualized information to search within.
>
> One potential solution is to first use RAG to retrieve documents relevant to the question, incorporate the retrieved text into the Induction-Gram context, and then allow Induction-Gram to generate accurate, interpretable outputs based on this enriched context. This approach could significantly improve prediction performance at the cost of some efficiency (as the retrieval step may be computationally expensive).
>
> **Q2: Assessment of Induction-Gram on Diverse Setting**
>
> We conducted two additional assessments to address the performance of Induction-Gram on domain-specific datasets and with larger reference corpora.
>
> **1) Domain-specific dataset**
>
> We evaluated the model using the Pile of Law dataset [a], which contains English-language legal and administrative texts, such as court opinions, contracts, administrative rules, and legislative records. The table below summarizes the next-token prediction performance on 50k samples from Pile of Law, using the LLaMA-2 tokenizer. For Infini-Gram, Pile-train was used as the reference corpus.
>
> | Models | Performance |
> |---|---|
> | Induction-only (exact) | 21.2 |
> | Induction-only (fuzzy) | 40.9 |
> | Infini-Gram | 54.8 |
> | Induction-Gram | 58.9 |
> | LLM(LLaMA2-7B) | 68.8 |
>
> These results demonstrate that Induction-Gram, with the introduction of induction-heads and fuzzy matching, outperforms Infini-Gram, consistent with the findings in our paper. Notably, Induction-Only (fuzzy) achieves significantly better performance than Induction-Only (exact). The complex and nuanced language in legal texts likely benefits more from the flexibility provided by fuzzy matching.
>
> **2) Impact of Larger Reference Corpora**
>
> Additionally, we explored the effect of using larger reference corpora on Induction-Gram’s performance. The table below presents next-token prediction accuracy on BabyLM-test (using the LLaMA-2 tokenizer) with an expanded reference corpus compared to the one used in the paper.
>
> | Reference Corpus (# of Tokens) | Infini-Gram | Induction-Gram | LLaMA-2-7B |
> |---|---|---|---|
> | RedPajama (1.39T) | 32.3 | 51.1 | 62.2 |
> | Dolma-v1.7 (2.60T) | 43.6 | 59.0 | 62.2 |
>
> These results highlight two key points:
> * Performance Scaling: The performance of Infini-Gram improves with a larger reference corpus, and this improvement translates directly to Induction-Gram.
> * Competitiveness with LLMs: Induction-Gram achieves performance close to that of LLaMA-2-7B, demonstrating its ability to benefit from larger datasets without compromising its interpretability.
>
> These findings demonstrate that our proposed strategies-*induction-heads* and *fuzzy matching*-remain effective even when scaling the corpus size, reinforcing their role in achieving the study’s objectives.
>
> **Q3: Impact of training data on the Fuzzy Matching Model**
>
> Regarding the training data for the Fuzzy Matching Model, we intentionally selected general datasets such as OpenWebText and Pile-train to ensure broad applicability. To explore the impact of different data distributions, we conducted an experiment using the Pile of Law dataset [a], which consists of English-language legal and administrative data, including court opinions, contracts, administrative rules, and legislative records. This dataset exhibits a significantly different distribution compared to OpenWebText and Pile-train.
>
> For a fair comparison, we trained the Fuzzy Matching Model on a similar number of samples from Pile of Law as we used for OpenWebText. The results are summarized in the table below. Models trained on Pile of Law showed slightly lower performance in Induction-Only (fuzzy) and Induction-Gram compared to those trained on OpenWebText. This suggests that the language style and topic distribution of the training dataset influence the learned similarity metrics in the Fuzzy Matching Model. That said, the performance difference is relatively small, approximately 1%p, indicating that while dataset selection impacts performance, the Fuzzy Matching Model and Induction-Gram remain robust across different training distributions.
>
> | Model  | OpenWebText | Pile of Law |
> |---|---|---|
> | Induction-only (fuzzy) | 43.2 | 41.8 |
> | Induction-Gram | 49.8 | 48.7 |
>
> [a] Henderson, Peter, et al. "Pile of law: Learning responsible data filtering from the law and a 256gb open-source legal dataset." Advances in Neural Information Processing Systems 35 (2022): 29217-29234.

---

### Official Review · Reviewer_W5e9 · 2024-11-04

**Soundness:** 2
**Presentation:** 3
**Contribution:** 3
**Rating:** 6
**Confidence:** 3

**Summary:**

This paper proposed a new language model formulation that integrates three distinct components: infini-gram constructed by a corpus, an induction-only model that searches for the input context and constructs a distribution over obtained subsequence, and induction-only model with fuzzy matching. These models are exclusively selected by the effective n (max length of matched sequence) so only one component is responsible to yield the distribution of the next token at each time.
Experiments show its accurate behavior especially when the effective n is large, as well as its compute efficiency when the proposed model is applied to speculative decoding. The paper also involves a downsteram evaluation on fMRI, showing the effectiveness in correlation against response fo brain voxels, and its qualitative summarization with LLMs.

**Strengths:**

As all the components in the proposed model rely on some sequences in the corpus and/or input context, the proposed model also maintains a certain transparency to calculate the word probability, though it doesn't guarantee the full interpretability on the result of fuzzy matching model.

**Weaknesses:**

The actual accuracy is far below the baseline LLM (gpt-2 and llama). In addition in Figure 4, any components of the proposed model don't surpass the baseline LLM (llama) when the effective n is small. This is one of the limitations of the proposed model that heavily relies on symbolic history.

Experiments in the Table 2 looks unfair. Only a bare Transformer is introduced as a baseline and any other compute-efficient models are not considered. The table also doesn't contain information about the statistics of proposed samples and their acceptance.

**Questions:**

It is still unclear about the actual relation between the proposed method and "induction head". I think the authors adopted this term for just an analogy, but I would expect some quantitative comparison between the existing analysis of "induction heads" and the result of this study.

More ablation study, especially investigating with one omission of the component is recommended. I got this curiosity because it looks the fuzzy model is generally better than the exact model except its compute efficiency.

It looks the fMRI experiment is interesting, but jsut superfluous and should be investigated in the another study. In contrast, there are few discussion about the behavior of the method itself in-depth in the paper. Because the paper targeted on the interpretability of the method, the authors should focus on reinforcing such information instead of spreading the interest of the paper.

---

> ### Author Response · Authors · 2024-11-22
>
> Thank you for your thoughtful observation and for recognizing the transparency of our proposed model. Below, we provide further clarification and present additional experimental results to address your concerns.  Some changes have also been incorporated into the revised manuscript, highlighted in blue for your convenience.
>
> **W1. Performance Compared to LLMs**
>
> Below, we clarify our study’s scope, objectives, and the relevance of our findings.
>
> **1) Scope and Objectives of the Study**
>
> As mentioned in the paper:
> > "... there is a considerable gap between the performance of interpretable models and black-box LLMs in next-token prediction."
>
> Our work aims to address this gap by improving the performance of interpretable models without sacrificing their transparency. Rather than directly competing with LLMs, our focus lies in enhancing interpretable language modeling. Specifically, we identified limitations of Infini-Gram, a state-of-the-art scalable n-gram model, such as difficulties in adapting to new contexts or handling mismatched queries (e.g., typos or rephrasings). To address these issues, we proposed two strategies: *induction-heads* and *fuzzy matching*. These techniques improve performance while preserving interpretability, as demonstrated throughout the paper.
> Thus, the significance of our work lies in demonstrating how transparent modeling can bridge the gap between interpretability and performance. By leveraging these strategies, Induction-Gram achieves substantial improvements over Infini-Gram and demonstrates the potential to approach the performance of black-box LLMs.
>
> **2) Impact of Larger Reference Corpora**
>
> Additionally, we explored the effect of using larger reference corpora on Induction-Gram’s performance. The table below presents next-token prediction accuracy on BabyLM-test (using the LLaMA-2 tokenizer) with an expanded reference corpus compared to the one used in the paper.
>
> | Reference Corpus (# of Tokens) | Infini-Gram | Induction-Gram | LLaMA-2-7B |
> |---|---|---|---|
> | RedPajama (1.39T) | 32.3 | 51.1 | 62.2 |
> | Dolma-v1.7 (2.60T) | 43.6 | 59.0 | 62.2 |
>
> These results highlight two key points:
> * Performance Scaling: The performance of Infini-Gram improves with a larger reference corpus, and this improvement translates directly to Induction-Gram.
> * Competitiveness with LLMs: Induction-Gram achieves performance close to that of LLaMA-2-7B, demonstrating its ability to benefit from larger datasets without compromising its interpretability.
>
> These findings demonstrate that our proposed strategies-*induction-heads* and *fuzzy matching*-remain effective even when scaling the corpus size, reinforcing their role in achieving the study’s objectives.
>
> **W2: Efficiency Comparison**
>
> We have included the acceptance rate in Table 2 to provide a more comprehensive evaluation.
> The updated table highlights that while the acceptance rate of our proposed method is lower compared to larger LLMs due to slightly reduced alignment with their outputs, it still achieves notable speed improvements during speculative decoding. This supports our primary objective of balancing speed and performance in practical, real-world applications.
>
> Regarding the baseline choice, we selected a bare Transformer to focus on the interpretability and impact of our specific contributions. While comparisons with other compute-efficient models, such as TinyLLaMA, are valuable, they fall beyond the scope of this work. That said, we have included additional comparisons with TinyLLaMA in an H100 environment in the revised version of the paper to further address this concern.

---

> > ### Author Response · Authors · 2024-11-22
> >
> > **Q1: Clarifying the Connection Between Induction Heads and Our Method**
> >
> > In LLMs, "induction heads" refer to a conceptual mechanism or role rather than a discrete module. As described in the literature [a]:
> >
> > > "Perhaps the most interesting finding was the induction head, a circuit whose function is to look back over the sequence for previous instances of the current token (call it A), find the token that came after it last time (call it B), and then predict that the same completion will occur again (e.g., forming the sequence [A][B] … [A] → [B]). In other words, induction heads 'complete the pattern' by copying and completing sequences that have occurred before."
> >
> > Our approach is inspired by this behavior and connects closely with the principles behind n-gram models, which also rely on patterns in past sequences to predict future tokens.
> >
> > That said, induction heads emerge as a distributed behavior across attention heads in LLMs, making a direct, quantitative comparison to n-gram models challenging. Instead of numerical results, we’ve included illustrative examples in Figure A2 of the Appendix to demonstrate how both exact and fuzzy matching in our method replicates the pattern-completion mechanism described for induction heads. These examples highlight how sequences from prior contexts are copied and extended, bridging the theoretical alignment. To address your concerns further, we’ve revised lines 180–182 of the manuscript for added clarity.
> >
> > [a] Olsson, Catherine, et al. "In-context learning and induction heads." arXiv preprint arXiv:2209.11895 (2022).
> >
> > **Q2: Ablation Study on the Impact of Components in Induction-Gram**
> >
> > We conducted an ablation study on Induction-Gram by systematically removing its individual components to evaluate their impact on next-token prediction. The results, presented in the table below, were assessed across multiple corpora of Infini-Gram, including BabyLM-dev, Pile-val, OpenWebText, and Pile-train.
> >
> > | Model | BabyLM-dev | Pile-val | OpenWebText | Pile-train |
> > |---|---|---|---|---|
> > | Induction-Gram | 43.1 | 42.9 | 43.2 | 49.4 |
> > | w/o  Induction-only (exact) | 43.0 | 42.8 | 43.1 | 49.3 |
> > | w/o  Infini-Gram | 42.9 | 42.9 | 42.9 | 42.9 |
> > | w/o  Induction-only (fuzzy) | 42.2 | 36.9 | 38.3 | 46.6 |
> > | Infini-Gram | 39.0 | 19.0 | 20.1 | 33.5 |
> >
> > The omission of fuzzy induction led to a notable performance drop, more significant than when exact induction was removed. This suggests that fuzzy matching is essential for enhancing the model's adaptability and overall performance, as further demonstrated in Table 1, which compares the results of Induction-only (exact) and Induction-only (fuzzy). Both exact induction and fuzzy induction serve complementary roles, much like "induction heads" in the model—when one is removed, the other partially compensates, leading to a smaller performance decline.
> >
> > Only when using Pile-train as a reference corpus, omitting \infinigram{} leads to the most substantial performance decline. Importantly, when the reference corpus does not align well with the test dataset (e.g., Pile-val, OpenWebText), Infini-Gram’s performance significantly worsens, falling below even the “w/o Infini-Gram” scenario. This underscores its sensitivity to the quality and relevance of the reference data.
> >
> > We have also included these results and detailed analysis in Section A3 of the Appendix.
> >
> > **Q3: Understanding the Role of fMRI Experiment in Induction-Gram Evaluation**
> >
> > While the neuroscience implications of the fMRI experiment may not be important to all ICLR readers, we believe its inclusion is crucial to (1) grounding the interpretability evaluation of Induction-Gram and (2) showing its generalizability across domains:
> >
> > **(1) Grounded interpretability evaluation**: Evaluating interpretability outside the context of a real problem is difficult and has been plagued by misleading results [b]. The fMRI experiment grounds our evaluation of interpretability in a real problem where interpretation is itself the end goal, rather than a contrived synthetic setting. Through our analysis in Section 5, we show how Induction-Gram can generate genuinely useful interpretations that go beyond existing methods. This serves as a case study for how the same analysis can be done in other domains where interpretability is critical (e.g., in analyzing medical notes or data in computational social science).
> >
> > **(2) Quantitative Evaluation**: While we focus on text-only models (text input-text output), the Induction-Gram methodology is quite general (it requires text input but can handle any sequential output). The fMRI experiment shows that this methodology significantly outperforms baseline interpretable models (Table 3). This consistency across domains highlights the generalizability of the Induction-Gram model.
> >
> > [b] “Sanity Checks for Saliency Maps” https://arxiv.org/abs/1810.03292

---

> ### Comment · Reviewer_W5e9 · 2024-12-03
>
> Thanks for providing thorough comments and additional information for my review. They cralified some of my questions and I would increase my recommendation by 1 as it sounds enough to express the advantage of the proposed method.
>
> I still recommend to split the fMRI part of the paper into another because it shrinks the space of the paper for analysing the method itself (e.g., I guessed the additional content in the appendix looks more reasonable to be included into the main text), but I wouldn't claim that this is critical.

---

### Official Review · Reviewer_uyGF · 2024-11-06

**Soundness:** 3
**Presentation:** 3
**Contribution:** 3
**Rating:** 8
**Confidence:** 2

**Summary:**

This paper improves on LLM interpretability by integrating a specially designed 'induction head' with state-of-the-art n-gram models. Beyond interpretability, authors present overall improvement in next-word prediction over baseline interpretable models.

**Strengths:**

* Completeness in presentation and experimentation
* Effective interpretable model with improvements in next word prediction and efficiency

**Weaknesses:**

* Not effective in all settings (e.g. short sentences or reasoning tasks)

**Questions:**

How will your conclusions be affected with larger models?

---

> ### Author Response · Authors · 2024-11-22
>
> We sincerely thank the reviewers for their constructive feedback and for acknowledging the effectiveness of our proposed model in next-token prediction. Your comments on the identified weaknesses and thoughtful questions have provided valuable directions for improvement. Below, we address each point in detail.
>
> **W1. Effectiveness in Various Settings**
>
> Our primary goal in this study is to improve upon the state-of-the-art for interpretable, next-token prediction (Infini-Gram) by adaptively incorporating input context. We show promising results in two domains: language modeling and fMRI encoding tasks. While we acknowledge the model’s constraints with shorter contexts, one potential direction could involve integrating retrieval-augmented generation (RAG) methods. This approach would allow Induction-Gram to retrieve and incorporate more representative texts from larger corpora, enhancing its contextual understanding. Although this might introduce additional computational steps, it could significantly improve accuracy and interpretability in short-context scenarios.
>
> For reasoning tasks, we agree that this remains a broader challenge, even for advanced black-box LLMs, which have yet to demonstrate consistent reasoning abilities [a, b, c]. Induction-Gram is designed to bridge interpretability and contextualization; advancing its reasoning capabilities goes beyond the scope of this paper. Future research could explore tailoring the model to address reasoning-specific challenges.
>
> [a] Wang, Siyuan, et al. "Can llms reason with rules? logic scaffolding for stress-testing and improving llms." arXiv preprint arXiv:2402.11442 (2024).
>
> [b] Valmeekam, Karthik, et al. "Large language models still can't plan (a benchmark for LLMs on planning and reasoning about change)." NeurIPS 2022 Foundation Models for Decision Making Workshop. 2022.
>
> [c] Li, Zhiming, et al. "Llms for relational reasoning: How far are we?." Proceedings of the 1st International Workshop on Large Language Models for Code. 2024.
>
>
> **Q1: Discussion on Larger Models**
>
> We interpret "larger models" in two possible ways: expanding the reference corpus for Infini-Gram or scaling up large language models (LLMs). Below, we address both interpretations.
>
> **1) Larger Reference Corpus for Infini-Gram**
>
> Infini-Gram serves as the foundation of our baseline, where scaling corresponds to increasing the size of the reference corpus. To explore this, we conducted additional experiments using a larger reference corpus released by the authors of Infini-Gram. The table below shows next-token prediction accuracy on BabyLM-test with the LLaMA-2 tokenizer.
>
> As seen in the results, Infini-Gram’s performance improves with a larger reference corpus, and Induction-Gram also shows a corresponding performance gain. This demonstrates that the inductive mechanism of Induction-Gram remains effective regardless of the size of the reference corpus, emphasizing its continued relevance even as Infini-Gram scales. Therefore, our conclusions remain consistent.
>
> | Reference Corpus (# of Tokens) | Infini-Gram | Induction-Gram |
> |---|---|---|
> | RedPajama (1.39T) | 32.3 | 51.1 |
> | Dolma-v1.7 (2.60T) | 43.6 | 59.0 |
>
> **2) Larger LLM**
>
> If the comment refers to comparisons with large-scale LLMs such as LLaMa3.1-405B, we omit these as they do not affect our main goal: improving the state-of-the-art for *interpretable* next-token prediction. Thus, we compare to SOTA baselines for interpretable next-token prediction, which still trail the black-box LLMs we compare against in performance. While larger LLMs often achieve superior raw performance, they do so at the expense of interpretability and computational cost. Our study prioritizes models that strike a balance between interpretability and performance. Induction-Gram excels in this niche, delivering meaningful performance improvements over other interpretable language models while maintaining a lightweight design.

---

### Author Response · Authors · 2024-12-03
**Final Response to Reviewers**

We sincerely thank all reviewers for their valuable feedback, which has greatly enhanced our work. In this paper, we introduce Induction-Gram, an efficient and interpretable language model that bridges the gap between traditional n-gram-based models and neural LLMs. Our key contributions include:
1. **Introducing Induction-Gram**: A novel approach integrating induction heads and fuzzy matching to address adaptability and query-matching limitations of n-gram models.
2. **Balancing Interpretability and Performance**: Demonstrating that interpretable models can achieve competitive performance without sacrificing transparency.
3. **fMRI Evaluation**: Grounding interpretability through real-world fMRI encoding tasks, showing generalizability across domains.
4. **Mechanistic Interpretability**: Advancing reverse-engineering of interpretable models from pre-trained LLMs, contributing to responsible AI development.

We have revised our manuscript based on reviewers’ comments and summarize our responses as follows:
* **Reviewer uyGF**: We discussed  Induction-Gram’s effectiveness in various settings (W1) and provided additional experimental results with larger models (Q1).
* **Reviewer W5e9**: We clarified our research goal (W1), the role and significance of fMRI evaluation (Q3), and the rationale for efficiency evaluation (W2). We expand on the concept of induction heads (Q1) and present results from ablation studies (Q2), which have been incorporated into the revised manuscript.
* **Reviewer YaGh**: We provided detailed discussions on the research goal and the rationale behind the comparisons made in our study (W1, Q2). Additional experiments, including an ablation study on the training of fuzzy matching models (W2, Q3) and assessments with new datasets and larger models (Q2), were included. We also elaborated on potential integrations with other methods, such as retrieval-augmented generation (Q1).
* **Reviewer RdyW**: We discussed the robustness of Induction-Gram across tokenization systems, highlighting consistent improvements regardless of tokenizer choice (W1).

We once again thank all reviewers for their insightful feedback and hope our responses address all concerns. Please feel free to reach out with any further questions or comments.

---

### Meta-Review · Area_Chair_swZb · 2024-12-19

**Metareview:**

The paper proposes a new approach called Induction-Gram, which integrates an "induction head" into n-gram models to improve interpretability and prediction efficiency. The method aims to bridge the gap between the transparency of traditional n-gram models and the high performance of modern LLMs. The induction head utilizes a custom neural similarity metric to identify potential next-word completions based on input context, significantly improving the accuracy of next-token predictions. The paper also explores the use of Induction-Gram for speculative decoding, offering potential performance benefits in large model inference. The model is additionally tested in the context of neuroscience, showing improved correlation with fMRI data in a task that involves predicting brain responses to language stimuli.

Strengths of the paper include its approach to interpretable language modeling integrating fuzzy matching for improved performance over n-gram models, and the potential applications in both language model applications and neuroscience. However, the paper also has notable weaknesses. One significant limitation is the weak performance of Induction-Gram compared to LLMs like GPT-2 and LLaMA2, particularly when the "effective n" is small, which diminishes the generalizability of the proposed approach. The paper fails to provide comprehensive baseline comparisons with other interpretable or compute-efficient models, which makes it difficult to assess its advantages. While the inclusion of fMRI experiments is novel, it shifts the focus of the paper and dilutes the analysis of the core method itself. There is also a lack of in-depth discussion of model scalability and performance with more diverse datasets or in complex linguistic settings.

The most important reason for rejecting the paper is the limited empirical evaluation, which primarily focuses on next-token prediction accuracy, which risks underestimating the performance gap, and lacks more comprehensive benchmarks such as perplexity or downstream task evaluations. The model’s performance does not sufficiently compete with existing LLMs, and the use of fMRI experiments, while interesting, feels tangential to the core contribution of the paper. Insufficient comparison with a broader range of models also raise concerns about the paper’s robustness. While the work presents an interesting idea, the execution and evaluation are not sufficient to justify acceptance for ICLR at this stage.

**Additional Comments On Reviewer Discussion:**

During the rebuttal period, several points were raised revolving around baseline comparisons, the inclusion of fMRI experiments, experimental clarity, and model performance.

Baseline Comparisons and Model Performance: Reviewer W5e9 and YaGh emphasized that the paper lacks sufficient baseline comparisons, particularly with other compute-efficient or interpretable models, making it difficult to assess the competitiveness of Induction-Gram. The authors responded by clarifying their focus on improving interpretable models rather than directly competing with black-box LLMs. They included additional comparisons with TinyLLaMA and expanded results using larger reference corpora.

Relevance of fMRI Experiments: Reviewer W5e9 and uyGF expressed concerns that the fMRI experiments, while novel, were tangential to the paper’s primary goal of improving interpretability in language models. Reviewer W5e9 suggested that the fMRI results should be split into a separate study to allow for more focus on the methodological contributions. In response, the authors defended the inclusion of the fMRI task, arguing that it grounds the interpretability evaluation in a real-world setting and highlights generalizability. While reviewers acknowledged this argument, the consensus was that it diluted the focus on the core contributions.

Experimental Clarity and Reproducibility: Reviewer YaGh and W5e9 flagged issues with the insufficient details of experimental settings, particularly for the fuzzy matching model and ablation studies. The authors addressed this by providing clarifications on hyperparameter settings, dataset choices, and additional ablation studies. Despite this, doubts remained about the model’s scalability and robustness across diverse datasets.

Performance in Shorter Contexts: Reviewer W5e9 noted that the model’s performance degrades significantly for shorter contexts, where the "effective n" is small, a key limitation of the proposed method. The authors acknowledged this and suggested that retrieval-augmented generation (RAG) could mitigate the issue. However, they admitted that this introduces additional computational costs, which remain a trade-off.

In weighing these points, I found that while the authors made some improvements and provided clarifications, the fundamental issues raised by reviewers persisted. The lack of comprehensive baseline comparisons, limited performance in shorter contexts, and the tangential inclusion of fMRI experiments ultimately weakened the paper’s overall contribution. The additional results, while helpful, were not sufficient to address concerns about the robustness and generalizability of the method. Although the reviewers acknowledged the novelty of the approach, the empirical evidence and focus of the work did not convincingly justify acceptance at ICLR.

---

### Decision · Program_Chairs · 2025-01-22

Reject